# Learning with Incremental Iterative Regularization

**Lorenzo Rosasco**
DIBRIS, Univ. Genova, ITALY
LCSL, IIT & MIT, USA
lrosasco@mit.edu

**Silvia Villa**
LCSL, IIT & MIT, USA
Silvia.Villa@iit.it

## Abstract

Within a statistical learning setting, we propose and study an iterative regularization algorithm for least squares defined by an incremental gradient method. In particular, we show that, if all other parameters are fixed a priori, the number of passes over the data (epochs) acts as a regularization parameter, and prove strong universal consistency, i.e. almost sure convergence of the risk, as well as sharp finite sample bounds for the iterates. Our results are a step towards understanding the effect of multiple epochs in stochastic gradient techniques in machine learning and rely on integrating statistical and optimization results.

## 1 Introduction

Machine learning applications often require efficient statistical procedures to process potentially massive amount of high dimensional data. Motivated by such applications, the broad objective of our study is designing learning procedures with optimal statistical properties, and, at the same time, computational complexities proportional to the *generalization* properties allowed by the data, rather than their raw amount [6]. We focus on iterative regularization as a viable approach towards this goal. The key observation behind these techniques is that iterative optimization schemes applied to scattered, noisy data exhibit a self-regularizing property, in the sense that early termination (early-stop) of the iterative process has a regularizing effect [21, 24]. Indeed, iterative regularization algorithms are classical in inverse problems [15], and have been recently considered in machine learning [36, 34, 3, 5, 9, 26], where they have been proved to achieve optimal learning bounds, matching those of variational regularization schemes such as Tikhonov [8, 31].

In this paper, we consider an iterative regularization algorithm for the square loss, based on a recursive procedure processing one training set point at each iteration. Methods of the latter form, often broadly referred to as online learning algorithms, have become standard in the processing of large data-sets, because of their low iteration cost and good practical performance. Theoretical studies for this class of algorithms have been developed within different frameworks. In composite and stochastic optimization [19, 20, 29], in online learning, a.k.a. sequential prediction [11], and finally, in statistical learning [10]. The latter is the setting of interest in this paper, where we aim at developing an analysis keeping into account simultaneously both statistical and computational aspects. To place our contribution in context, it is useful to emphasize the role of regularization and different ways in which it can be incorporated in online learning algorithms. The key idea of regularization is that controlling the *complexity* of a solution can help avoiding overfitting and ensure stability and generalization [33]. Classically, regularization is achieved penalizing the objective function with some suitable functional, or minimizing the risk on a restricted space of possible solutions [33]. Model selection is then performed to determine the amount of regularization suitable for the data at hand. More recently, there has been an interest in alternative, possibly more efficient, ways to incorporate regularization. We mention in particular [1, 35, 32] where there is no explicit regularization by penalization, and the step-size of an iterative procedure is shown to act as a regularization parameter. Here, for each fixed step-size, each data point is processed once, but multiple passes are typically needed to perform model selection (that is, to pick the best step-size). We also mention

[22] where an interesting adaptive approach is proposed, which seemingly avoid model selection under certain assumptions.

In this paper, we consider a different regularization strategy, widely used in practice. Namely, we consider no explicit penalization, fix the step size a priori, and analyze the effect of the number of passes over the data, which becomes the only free parameter to avoid overfitting, i.e. regularize. The associated regularization strategy, that we dub *incremental iterative regularization*, is hence based on early stopping. The latter is a well known "trick", for example in training large neural networks [18], and is known to perform very well in practice [16]. Interestingly, early stopping with the square loss has been shown to be related to boosting [7], see also [2, 17, 36]. Our goal here is to provide a theoretical understanding of the generalization property of the above heuristic for incremental/online techniques. Towards this end, we analyze the behavior of both the excess risk and the iterates themselves. For the latter we obtain sharp finite sample bounds matching those for Tikhonov regularization in the same setting. Universal consistency and finite sample bounds for the excess risk can then be easily derived, albeit possibly suboptimal. Our results are developed in a capacity independent setting [12, 30], that is under no conditions on the covering or entropy numbers [30]. In this sense our analysis is worst case and dimension free. To the best of our knowledge the analysis in the paper is the first theoretical study of regularization by early stopping in incremental/online algorithms, and thus a first step towards understanding the effect of multiple passes of stochastic gradient for risk minimization.

The rest of the paper is organized as follows. In Section 2 we describe the setting and the main assumptions, and in Section 3 we state the main results, discuss them and provide the main elements of the proof, which is deferred to the supplementary material. In Section 4 we present some experimental results on real and synthetic datasets.

**Notation** We denote by $\mathbb{R}_+ = [0, +\infty[$, $\mathbb{R}_{++} = ]0, +\infty[$, and $\mathbb{N}^* = \mathbb{N} \setminus \{0\}$. Given a normed space $\mathcal{B}$ and linear operators $(A_i)_{1 \leq i \leq m}$, $A_i \colon \mathcal{B} \to \mathcal{B}$ for every $i$, their composition $A_m \circ \cdots \circ A_1$ will be denoted as $\prod_{i=1}^{m} A_i$. By convention, if $j > m$, we set $\prod_{i=j}^{m} A_i = I$, where $I$ is the identity of $\mathcal{B}$. The operator norm will be denoted by $\| \cdot \|$ and the Hilbert-Schmidt norm by $\| \cdot \|_{HS}$. Also, if $j > m$, we set $\sum_{i=j}^{m} A_i = 0$.

## 2 Setting and Assumptions

We first describe the setting we consider, and then introduce and discuss the main assumptions that will hold throughout the paper. We build on ideas proposed in [13, 27] and further developed in a series of follow up works [8, 3, 28, 9]. Unlike these papers where a reproducing kernel Hilbert space (RKHS) setting is considered, here we consider a formulation within an abstract Hilbert space. As discussed in the Appendix A, results in a RKHS can be recovered as a special case. The formulation we consider is close to the setting of functional regression [25] and reduces to standard linear regression if $\mathcal{H}$ is finite dimensional, see Appendix A.

Let $\mathcal{H}$ be a separable Hilbert space with inner product and norm denoted by $\langle \cdot, \cdot \rangle_{\mathcal{H}}$ and $\| \cdot \|_{\mathcal{H}}$. Let $(X, Y)$ be a pair of random variables on a probability space $(\Omega, \mathfrak{S}, \mathbb{P})$, with values in $\mathcal{H}$ and $\mathbb{R}$, respectively. Denote by $\rho$ the distribution of $(X, Y)$, by $\rho_X$ the marginal measure on $\mathcal{H}$, and by $\rho(\cdot | x)$ the conditional measure on $\mathbb{R}$ given $x \in \mathcal{H}$. Considering the square loss function, the problem under study is the minimizazion of the *risk*,

$$\inf_{w \in \mathcal{H}} \mathcal{E}(w), \quad \mathcal{E}(w) = \int_{\mathcal{H} \times \mathbb{R}} (\langle w, x \rangle_{\mathcal{H}} - y)^2 d\rho(x, y), \tag{1}$$

provided the distribution $\rho$ is fixed but known only through a *training set* $\mathbf{z} = \{(x_1, y_1), \ldots, (x_n, y_n)\}$, that is a realization of $n \in \mathbb{N}^*$ independent identical copies of $(X, Y)$. In the following, we measure the quality of an approximate solution $\hat{w} \in \mathcal{H}$ (an estimator) considering the *excess risk*

$$\mathcal{E}(\hat{w}) - \inf_{\mathcal{H}} \mathcal{E}. \tag{2}$$

If the set of solutions of Problem (1) is non empty, that is $\mathcal{O} = \operatorname{argmin}_{\mathcal{H}} \mathcal{E} \neq \varnothing$, we also consider

$$\left\| \hat{w} - w^\dagger \right\|_{\mathcal{H}}, \quad \text{where} \quad w^\dagger = \operatorname*{argmin}_{w \in \mathcal{O}} \|w\|_{\mathcal{H}}. \tag{3}$$

More precisely we are interested in deriving almost sure convergence results and finite sample bounds on the above error measures. This requires making some assumptions that we discuss next. We make throughout the following basic assumption.

**Assumption 1.** *There exist $M \in ]0, +\infty[$ and $\kappa \in ]0, +\infty[$ such that $|y| \leq M$ $\rho$-almost surely, and $\|x\|_{\mathcal{H}}^2 \leq \kappa$ $\rho_X$-almost surely.*

The above assumption is fairly standard. The boundness assumption on the output is satisfied in classification, see Appendix A, and can be easily relaxed, see e.g. [8]. The boundness assumption on the input can also be relaxed, but the resulting analysis is more involved. We omit these developments for the sake of clarity. It is well known that (see e.g. [14]), under Assumption 1, the risk is a convex and continuous functional on $L^2(\mathcal{H}, \rho_X)$, the space of square-integrable functions with norm $\|f\|_{\rho}^2 = \int_{\mathcal{H} \times \mathbb{R}} |f(x)|^2 d\rho_X(x)$. The minimizer of the risk on $L^2(\mathcal{H}, \rho_X)$ is the regression function $f_\rho(x) = \int y d\rho(y|x)$ for $\rho_X$-almost every $x \in \mathcal{H}$. By considering Problem (1) we are restricting the search for a solution to linear functions. Note that, since $\mathcal{H}$ is in general infinite dimensional, the minimum in (1) might not be achieved. Indeed, bounds on the error measures in (2) and (3) depend on if, and how well, the regression function can be linearly approximated. The following assumption quantifies in a precise way such a requirement.

**Assumption 2.** *Consider the space $\mathcal{L}_\rho = \{f : \mathcal{H} \to \mathbb{R} \mid \exists w \in \mathcal{H} \text{ with } f(x) = \langle w, x \rangle \ \rho_X\text{- a.s.}\}$, and let $\overline{\mathcal{L}_\rho}$ be its closure in $L^2(\mathcal{H}, \rho_X)$. Moreover, consider the operator*

$$L : L^2(\mathcal{H}, \rho_X) \to L^2(\mathcal{H}, \rho_X), \quad Lf(x) = \int_{\mathcal{H}} \langle x, x' \rangle f(x') d\rho(x'), \quad \forall f \in L^2(\mathcal{H}, \rho_X). \quad (4)$$

*Define $g_\rho = \operatorname{argmin}_{g \in \overline{\mathcal{L}_\rho}} \|f_\rho - g\|_\rho$. Let $r \in [0, +\infty[$, and assume that*

$$(\exists g \in L^2(\mathcal{H}, \rho_X)) \quad \text{such that} \quad g_\rho = L^r g. \quad (5)$$

The above assumption is standard in the context of RKHS [8]. Since its statement is somewhat technical, and we provide a formulation in a Hilbert space with respect to the usual RKHS setting, we further comment on its interpretation. We begin noting that $\mathcal{L}_\rho$ is the space of linear functions indexed by $\mathcal{H}$ and is a proper subspace of $L^2(\mathcal{H}, \rho_X)$ – if Assumption 1 holds. Moreover, under the same assumption, it is easy to see that the operator $L$ is linear, self-adjoint, positive definite and trace class, hence compact, so that its fractional power in (4) is well defined. Most importantly, the following equality, which is analogous to Mercer's theorem [30], can be shown fairly easily:

$$\mathcal{L}_\rho = L^{1/2} \left( L^2(\mathcal{H}, \rho_X) \right). \quad (6)$$

This last observation allows to provide an interpretation of Condition (5). Indeed, given (6), for $r = 1/2$, Condition (5) states that $g_\rho$ belongs to $\mathcal{L}_\rho$, rather than its closure. In this case, Problem 1 has at least one solution, and the set $\mathcal{O}$ in (3) is not empty. Vice versa, if $\mathcal{O} \neq \varnothing$ then $g_\rho \in \mathcal{L}_\rho$, and $w^\dagger$ is well-defined. If $r > 1/2$ the condition is stronger than for $r = 1/2$, for the subspaces of $L^r(L^2(\mathcal{H}, \rho_X))$ are nested subspaces of $L^2(\mathcal{H}, \rho_X)$ for increasing $r$[1].

## 2.1 Iterative Incremental Regularized Learning

The learning algorithm we consider is defined by the following iteration.

Let $\hat{w}_0 \in \mathcal{H}$ and $\gamma \in \mathbb{R}_{++}$. Consider the sequence $(\hat{w}_t)_{t \in \mathbb{N}}$ generated through the following procedure: given $t \in \mathbb{N}$, define

$$\hat{w}_{t+1} = \hat{u}_t^n, \quad (7)$$

where $\hat{u}_t^n$ is obtained at the end of one cycle, namely as the last step of the recursion

$$\hat{u}_t^0 = \hat{w}_t; \qquad \hat{u}_t^i = \hat{u}_t^{i-1} - \frac{\gamma}{n}(\langle \hat{u}_t^{i-1}, x_i \rangle_{\mathcal{H}} - y_i)x_i, \quad i = 1, \dots, n. \quad (8)$$

Each cycle, called an epoch, corresponds to one pass over data. The above iteration can be seen as the incremental gradient method [4, 19] for the minimization of the empirical risk corresponding to **z**, that is the functional,

$$\hat{\mathcal{E}}(w) = \frac{1}{n} \sum_{i=1}^{n} (\langle w, x_i \rangle_{\mathcal{H}} - y_i)^2. \tag{9}$$

(see also Section B.2). Indeed, there is a vast literature on how the iterations (7), (8) can be used to minimize the empirical risk [4, 19]. Unlike these studies in this paper we are interested in how the iterations (7), (8) can be used to approximately minimize the risk $\mathcal{E}$. The key idea is that while $\hat{w}_t$ is close to a minimizer of the empirical risk when $t$ is sufficiently large, a good approximate solution of Problem (1) can be found by terminating the iterations earlier (early stopping). The analysis in the next few sections grounds theoretically this latter intuition.

**Remark 1** (Representer theorem). *Let $\mathcal{H}$ be a RKHS of functions from $\mathcal{X}$ to $\mathcal{Y}$ defined by a kernel $K \colon \mathcal{X} \times \mathcal{X} \to \mathbb{R}$. Let $\hat{w}_0 = 0$, then the iteration after $t$ epochs of the algorithm in (7)-(8) can be written as $\hat{w}_t(\cdot) = \sum_{k=1}^{n} (\alpha_t)_k K_{x_k}$, for suitable coefficients $\alpha_t = ((\alpha_t)_1, \ldots, (\alpha_t)_n) \in \mathbb{R}^n$, updated as follows:*

$$\alpha_{t+1} = c_t^n$$

$$c_t^0 = \alpha_t, \quad (c_t^i)_k = \begin{cases} (c_t^{i-1})_k - \dfrac{\gamma}{n} \left( \sum_{j=1}^{n} K(x_i, x_j)(c_t^{i-1})_j - y_i \right), & k = i \\ (c_t^{i-1})_k, & k \neq i \end{cases}$$

## 3 Early stopping for incremental iterative regularization

In this section, we present and discuss the main results of the paper, together with a sketch of the proof. The complete proofs can be found in Appendix B. We first present convergence results and then finite sample bounds for the quantities in (2) and (3).

**Theorem 1.** *In the setting of Section 2, let Assumption 1 hold. Let $\gamma \in \left]0, \kappa^{-1}\right]$. Then the following hold:*

(i) *If we choose a stopping rule $t^* \colon \mathbb{N}^* \to \mathbb{N}^*$ such that*

$$\lim_{n \to +\infty} t^*(n) = +\infty \quad and \quad \lim_{n \to +\infty} \frac{t^*(n)^3 \log n}{n} = 0 \tag{10}$$

*then*

$$\lim_{n \to +\infty} \mathcal{E}(\hat{w}_{t^*(n)}) - \inf_{w \in \mathcal{H}} \mathcal{E}(w) = 0 \quad \mathbb{P}\text{-almost surely.} \tag{11}$$

(ii) *Suppose additionally that the set $\mathcal{O}$ of minimizers of (1) is nonempty and let $w^\dagger$ be defined as in (3). If we choose a stopping rule $t^* \colon \mathbb{N}^* \to \mathbb{N}^*$ satisfying the conditions in (10) then*

$$\|\hat{w}_{t^*(n)} - w^\dagger\|_{\mathcal{H}} \to 0 \quad \mathbb{P}\text{-almost surely.} \tag{12}$$

The above result shows that for an a priori fixed step-sized, consistency is achieved computing a suitable number $t^*(n)$ of iterations of algorithm (7)-(8) given $n$ points. The number of required iterations tends to infinity as the number of available training points increases. Condition (10) can be interpreted as an early stopping rule, since it requires the number of epochs not to grow too fast. In particular, this excludes the choice $t^*(n) = 1$ for all $n \in \mathbb{N}^*$, namely considering only one pass over the data. In the following remark we show that, if we let $\gamma = \gamma(n)$ to depend on the length of one epoch, convergence is recovered also for one pass.

**Remark 2** (Recovering Stochastic Gradient descent). *Algorithm in (7)-(8) for $t = 1$ is a stochastic gradient descent (one pass over a sequence of i.i.d. data) with stepsize $\gamma/n$. Choosing $\gamma(n) = \kappa^{-1} n^\alpha$, with $\alpha < 1/5$ in Algorithm (7)-(8), we can derive almost sure convergence of $\mathcal{E}(\hat{w}_1) - \inf_{\mathcal{H}} \mathcal{E}$ as $n \to +\infty$ relying on a similar proof to that of Theorem 1.*

To derive finite sample bounds further assumptions are needed. Indeed, we will see that the behavior of the bias of the estimator depends on the smoothness Assumption 2. We are in position to state our main result, giving a finite sample bound.

**Theorem 2** (Finite sample bounds in $\mathcal{H}$). *In the setting of Section 2, let $\gamma \in \left]0, \kappa^{-1}\right]$ for every $t \in \mathbb{N}$. Suppose that Assumption 2 is satisfied for some $r \in \left]1/2, +\infty\right[$. Then the set $\mathcal{O}$ of minimizers of* (1) *is nonempty, and $w^\dagger$ in* (3) *is well defined. Moreover, the following hold:*

(i) *There exists $c \in \left]0, +\infty\right[$ such that, for every $t \in \mathbb{N}^*$, with probability greater than $1 - \delta$,*

$$\|\hat{w}_t - w^\dagger\|_{\mathcal{H}} \leq \frac{32 \log \frac{16}{\delta}}{\sqrt{n}} \left( M\kappa^{-1/2} + 2M^2\kappa^{-1} + 3\|g\|_\rho \kappa^{r-\frac{3}{2}} \right) t + \left( \frac{r - \frac{1}{2}}{\gamma} \right)^{r-\frac{1}{2}} \|g\|_\rho t^{\frac{1}{2}-r}. \quad (13)$$

(ii) *For the stopping rule $t^* : \mathbb{N}^* \to \mathbb{N}^* : t^*(n) = \left\lceil n^{\frac{1}{2r+1}} \right\rceil$, with probability greater than $1 - \delta$,*

$$\|\hat{w}_{t^*(n)} - w^\dagger\|_{\mathcal{H}} \leq \left[ 32 \log \frac{16}{\delta} \left( M\kappa^{-1/2} + 2M^2\kappa^{-1} + 3\|g\|_\rho \kappa^{r-\frac{3}{2}} \right) + \left( \frac{r - \frac{1}{2}}{\gamma} \right)^{r-\frac{1}{2}} \|g\|_\rho \right] n^{-\frac{r-\frac{1}{2}}{2r+1}}. \quad (14)$$

The dependence on $\kappa$ suggests that a big $\kappa$, which corresponds to a small $\gamma$, helps in decreasing the sample error, but increases the approximation error. Next we present the result for the excess risk. We consider only the attainable case, that is the case $r > 1/2$ in Assumption 2. The case $r \leq 1/2$ is deferred to Appendix A, since both the proof and the statement are conceptually similar to the attainable case.

**Theorem 3** (Finite sample bounds for the risk – attainable case). *In the setting of Section 2, let Assumptions 1 holds, and let $\gamma \in \left]0, \kappa^{-1}\right]$. Let Assumption 2 be satisfied for some $r \in \left]1/2, +\infty\right]$. Then the following hold:*

(i) *For every $t \in \mathbb{N}^*$, with probability greater than $1 - \delta$,*

$$\mathcal{E}(\hat{w}_t) - \inf_{\mathcal{H}} \mathcal{E} \leq \frac{2\left(32 \log(16/\delta)\right)^2}{n} \left[ M + 2M^2\kappa^{-1/2} + 3\kappa^r \|g\|_\rho \right]^2 t^2 + 2\left( \frac{r}{\gamma t} \right)^{2r} \|g\|_\rho^2 \quad (15)$$

(ii) *For the stopping rule $t^* : \mathbb{N}^* \to \mathbb{N}^* : t^*(n) = \left\lceil n^{\frac{1}{2(1+r)}} \right\rceil$, with probability greater than $1 - \delta$,*

$$\mathcal{E}(\hat{w}_{t^*(n)}) - \inf_{\mathcal{H}} \mathcal{E} \leq \left[ 8 \left( 32 \log \frac{16}{\delta} \right)^2 \left( M + 2M^2\kappa^{-1/2} + 3\kappa^r \|g\|_\rho \right)^2 + 2\left( \frac{r}{\gamma} \right)^{2r} \|g\|_\rho^2 \right] n^{-r/(r+1)} \quad (16)$$

Equations (13) and (15) arise from a form of bias-variance (sample-approximation) decomposition of the error. Choosing the number of epochs that optimize the bounds in (13) and (15) we derive a priori stopping rules and corresponding bounds (14) and (16). Again, these results confirm that the number of epochs acts as a regularization parameter and the best choices following from equations (13) and (15) suggest multiple passes over the data to be beneficial. In both cases, the stopping rule depends on the smoothness parameter $r$ which is typically unknown, and hold-out cross validation is often used in practice. Following [9], it is possible to show that this procedure allows to adaptively achieve the same convergence rate as in (16).

## 3.1 Discussion

In Theorem 2, the obtained bound can be compared to known lower bounds, as well as to previous results for least squares algorithms obtained under Assumption 2. Minimax lower bounds and individual lower bounds [8, 31], suggest that, for $r > 1/2$, $O(n^{(r-1/2)/(2r+1)})$ is the optimal capacity-independent bound for the $\mathcal{H}$ norm[2]. In this sense, Theorem 2 provides sharp bounds on the iterates. Bounds can be improved only under stronger assumptions, e.g. on the covering numbers or on the eigenvalues of $L$ [30]. This question is left for future work. The lower bounds for the excess risk [8, 31] are of the form $O(n^{-2r/(2r+1)})$ and in this case the results in Theorems 3 and 7 are not sharp. Our results can be contrasted with online learning algorithms that use step-size

as regularization parameter. Optimal capacity independent bounds are obtained in [35], see also [32] and indeed such results can be further improved considering capacity assumptions, see [1] and references therein. Interestingly, our results can also be contrasted with non incremental iterative regularization approaches [36, 34, 3, 5, 9, 26]. Our results show that incremental iterative regularization, with distribution independent step-size, behaves as a batch gradient descent, at least in terms of iterates convergence. Proving advantages of incremental regularization over the batch one is an interesting future research direction. Finally, we note that optimal capacity independent and dependent bounds are known for several least squares algorithms, including Tikhonov regularization, see e.g. [31], and spectral filtering methods [3, 9]. These algorithms are essentially equivalent from a statistical perspective but different from a computational perspective.

## 3.2 Elements of the proof

The proofs of the main results are based on a suitable decomposition of the error to be estimated as the sum of two quantities that can be interpreted as a sample and an approximation error, respectively. Bounds on these two terms are then provided. The main technical contribution of the paper is the sample error bound. The difficulty in proving this result is due to multiple passes over the data, which induce statistical dependencies in the iterates.

**Error decomposition.** We consider an auxiliary iteration $(w_t)_{t\in\mathbb{N}}$ which is the expectation of the iterations (7) and (8), starting from $w_0 \in \mathcal{H}$ with step-size $\gamma \in \mathbb{R}_{++}$. More explicitly, the considered iteration generates $w_{t+1}$ according to

$$w_{t+1} = u_t^n, \tag{17}$$

where $u_t^n$ is given by

$$u_t^0 = w_t; \qquad u_t^i = u_t^{i-1} - \frac{\gamma}{n}\int_{\mathcal{H}\times\mathbb{R}}\left(\langle u_t^{i-1}, x\rangle_{\mathcal{H}} - y\right) x\, d\rho(x,y). \tag{18}$$

If we let $S\colon \mathcal{H} \to L^2(\mathcal{H}, \rho_X)$ be the linear map $w \mapsto \langle w, \cdot\rangle_{\mathcal{H}}$, which is bounded by $\sqrt{\kappa}$ under Assumption 1, then it is well-known that [13]

$$(\forall t \in \mathbb{N}) \quad \mathcal{E}(\hat{w}_t) - \inf_{\mathcal{H}}\mathcal{E} = \|S\hat{w}_t - g_\rho\|_\rho^2 \leq 2\|S\hat{w}_t - Sw_t\|_\rho^2 + 2\|Sw_t - g_\rho\|_\rho^2$$

$$\leq 2\kappa\|\hat{w}_t - w_t\|_{\mathcal{H}}^2 + 2(\mathcal{E}(w_t) - \inf_{\mathcal{H}}\mathcal{E}). \tag{19}$$

In this paper, we refer to the gap between the empirical and the expected iterates $\|\hat{w}_t - w_t\|_{\mathcal{H}}$ as the *sample error*, and to $\mathcal{A}(t, \gamma, n) = \mathcal{E}(w_t) - \inf_{\mathcal{H}}\mathcal{E}$ as the *approximation error*. Similarly, if $w^\dagger$ (as defined in (3)) exists, using the triangle inequality, we obtain

$$\|\hat{w}_t - w^\dagger\|_{\mathcal{H}} \leq \|\hat{w}_t - w_t\|_{\mathcal{H}} + \|w_t - w^\dagger\|_{\mathcal{H}}. \tag{20}$$

**Proof main steps.** In the setting of Section 2, we summarize the key steps to derive a general bound for the sample error (the proof of the behavior of the approximation error is more standard). The bound on the sample error is derived through many technical lemmas and uses concentration inequalities applied to martingales (the crucial point is the inequality in **STEP 5** below). Its complete derivation is reported in Appendix B.2. We introduce the additional linear operators: $T\colon \mathcal{H} \to \mathcal{H}\colon T = S^*S$, and, for every $x \in \mathcal{X}$, $S_x\colon \mathcal{H} \to \mathbb{R}\colon S_x w = \langle w, x\rangle$, and $T_x\colon \mathcal{H} \to \mathcal{H}\colon T_x = S_x S_x^*$. Moreover, set $\hat{T} = \sum_{i=1}^n T_{x_i}/n$. We are now ready to state the main steps of the proof.
**Sample error bound (STEP 1 to 5)**
**STEP 1 (see Proposition 1):** Find equivalent formulations for the sequences $\hat{w}_t$ and $w_t$:

$$\hat{w}_{t+1} = (I - \gamma\hat{T})\hat{w}_t + \gamma\left(\frac{1}{n}\sum_{j=1}^n S_{x_j}^* y_j\right) + \gamma^2\left(\hat{A}\hat{w}_t - \hat{b}\right)$$

$$w_{t+1} = (I - \gamma T)w_t + \gamma S^* g_\rho + \gamma^2(Aw_t - b),$$

where

$$\hat{A} = \frac{1}{n^2} \sum_{k=2}^{n} \left[ \prod_{i=k+1}^{n} \left( I - \frac{\gamma}{n} T_{x_i} \right) \right] T_{x_k} \sum_{j=1}^{k-1} T_{x_j}, \quad \hat{b} = \frac{1}{n^2} \sum_{k=2}^{n} \left[ \prod_{i=k+1}^{n} \left( I - \frac{\gamma}{n} T_{x_i} \right) \right] T_{x_k} \sum_{j=1}^{k-1} S_{x_j}^* y_j.$$

$$A = \frac{1}{n^2} \sum_{k=2}^{n} \left[ \prod_{i=k+1}^{n} \left( I - \frac{\gamma}{n} T \right) \right] T \sum_{j=1}^{k-1} T, \qquad b = \frac{1}{n^2} \sum_{k=2}^{n} \left[ \prod_{i=k+1}^{n} \left( I - \frac{\gamma}{n} T \right) \right] T \sum_{j=1}^{k-1} S^* g_\rho.$$

**STEP 2 (see Lemma 5):** Use the formulation obtained in **STEP 1** to derive the following recursive inequality,

$$\hat{w}_t - w_t = \left( I - \gamma \hat{T} + \gamma^2 \hat{A} \right)^t (\hat{w}_0 - w_0) + \gamma \sum_{k=0}^{t-1} \left( I - \gamma \hat{T} + \gamma \hat{A} \right)^{t-k+1} \zeta_k$$

with $\zeta_k = (T - \hat{T})w_k + \gamma(\hat{A} - A)w_k + \left( \frac{1}{n} \sum_{i=1}^{n} \hat{S}_{x_i}^* y_i - S^* g_\rho \right) + \gamma(b - \hat{b})$.

**STEP 3 (see Lemmas 6 and 7):** Initialize $\hat{w}_0 = w_0 = 0$, prove that $\|I - \gamma \hat{T} + \gamma \hat{A}\| \leq 1$, and derive from **STEP 2** that,

$$\|\hat{w}_t - w_t\|_{\mathcal{H}} \leq \gamma \big( \|T - \hat{T}\| + \gamma \|\hat{A} - A\| \big) \sum_{k=0}^{t-1} \|w_k\|_{\mathcal{H}} + \gamma t \Big( \Big\| \frac{1}{n} \sum_{i=1}^{n} \hat{S}_{x_i}^* y_i - S^* g_\rho \Big\| + \gamma \|b - \hat{b}\| \Big).$$

**STEP 4 (see Lemma 8):** Let Assumption 2 hold for some $r \in \mathbb{R}_+$ and $g \in L^2(\mathcal{H}, \rho_X)$. Prove that

$$(\forall t \in \mathcal{H}) \quad \|w_t\|_{\mathcal{H}} \leq \begin{cases} \max\{\kappa^{r-1/2}, (\gamma t)^{1/2-r}\} \|g\|_\rho & \text{if } r \in [0, 1/2[, \\ \kappa^{r-1/2} \|g\|_\rho & \text{if } r \in [1/2, +\infty[ \end{cases}$$

**STEP 5 (see Lemma 9 and Proposition 2:** Prove that with probability greater than $1 - \delta$ the following inequalities hold:

$$\|\hat{A} - A\|_{HS} \leq \frac{32\kappa^2}{3\sqrt{n}} \log \frac{4}{\delta}, \qquad \|\hat{b} - b\|_{\mathcal{H}} \leq \frac{32\kappa M^2}{3\sqrt{n}} \log \frac{4}{\delta},$$

$$\Big\| \hat{T} - T \Big\|_{HS} \leq \frac{16\kappa}{3\sqrt{n}} \log \frac{2}{\delta}, \qquad \Big\| \frac{1}{n} \sum_{i=1}^{n} S_{x_i}^* y_i - S^* g_\rho \Big\|_{\mathcal{H}} \leq \frac{16\sqrt{\kappa} M}{3\sqrt{n}} \log \frac{2}{\delta}.$$

**STEP 6 (approximation error bound, see Theorem 6):** Prove that, if Assumption 2 holds for some $r \in ]0, +\infty[$, then $\mathcal{E}(w_t) - \inf_{\mathcal{H}} \mathcal{E} \leq \left( r/\gamma t \right)^{2r} \|g\|_\rho^2$. Moreover, if Assumption 2 holds with $r = 1/2$, then $\|w_t - w^\dagger\|_{\mathcal{H}} \to 0$, and if Assumption 2 holds for some $r \in ]1/2, +\infty[$, then $\|w_t - w^\dagger\|_{\mathcal{H}} \leq \left( \frac{r-1/2}{\gamma t} \right)^{r-1/2} \|g\|_\rho$.

**STEP 7:** Plug the sample and approximation error bounds obtained in **STEP 1-5** and **STEP 6** in (19) and (20), respectively.

## 4 Experiments

**Synthetic data.** We consider a scalar linear regression problem with random design. The input points $(x_i)_{1 \leq i \leq n}$ are uniformly distributed in $[0, 1]$ and the output points are obtained as $y_i = \langle w^*, \Phi(x_i) \rangle + N_i$, where $N_i$ is a Gaussian noise with zero mean and standard deviation 1 and $\Phi = (\varphi_k)_{1 \leq k \leq d}$ is a dictionary of functions whose $k$-th element is $\varphi_k(x) = \cos((k-1)x) + \sin((k-1)x)$. In Figure 1, we plot the test error for $d = 5$ (with $n = 80$ in (a) and 800 in (b)). The plots show that the number of the epochs acts as a regularization parameter, and that early stopping is beneficial to achieve a better test error. Moreover, according to the theory, the experiments suggest that the number of performed epochs increases if the number of available training points increases.

**Real data.** We tested the kernelized version of our algorithm (see Remark 1 and Appendix A) on the cpuSmall[3], Adult and Breast Cancer Wisconsin (Diagnostic)[4] real-world

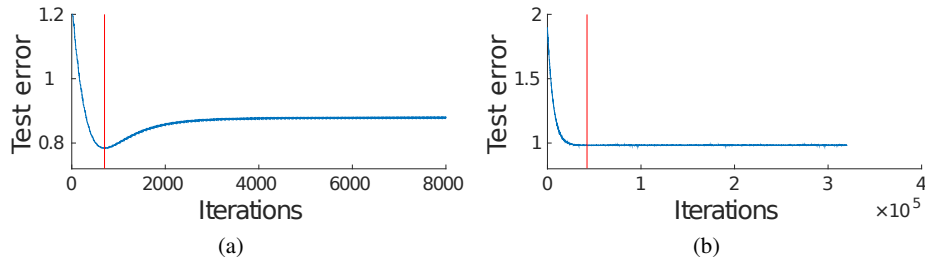

(a)                                    (b)

Figure 1: Test error as a function of the number of iterations. In (a), $n = 80$, and total number of iterations of IIR is 8000, corresponding to 100 epochs. In (b), $n = 800$ and the total number of epochs is 400. The best test error is obtained for 9 epochs in (a) and for 31 epochs in (b).

datasets. We considered a subset of `Adult`, with $n = 1600$. The results are shown in Figure 2. A comparison of the test errors obtained with the kernelized version of the method proposed in this paper (Kernel Incremental Iterative Regularization (KIIR)), Kernel Iterative Regularization (KIR), that is the kernelized version of gradient descent, and Kernel Ridge Regression (KRR) is reported in Table 1. The results show that the test error of KIIR is comparable to that of KIR and KRR.

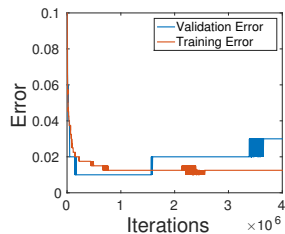

Figure 2: Training (orange) and validation (blue) classification errors obtained by KIIR on the `Breast Cancer` dataset as a function of the number of iterations. The test error increases after a certain number of iterations, while the training error is "decreasing" with the number of iterations.

Table 1: Test error comparison on real datasets. Median values over 5 trials.

| Dataset | $n_{tr}$ | d | Error Measure | KIIR | KRR | KIR |
|---|---|---|---|---|---|---|
| cpuSmall | 5243 | 12 | RMSE | 5.9125 | 3.6841 | 5.4665 |
| Adult | 1600 | 123 | Class. Err. | 0.167 | 0.164 | 0.154 |
| Breast Cancer | 400 | 30 | Class. Err. | 0.0118 | 0.0118 | 0.0237 |

## Acknowledgments

This material is based upon work supported by CBMM, funded by NSF STC award CCF-1231216. and by the MIUR FIRB project RBFR12M3AC. S. Villa is member of GNAMPA of the Istituto Nazionale di Alta Matematica (INdAM).

## Footnotes

[1]If $r < 1/2$ then the regression function does not have a best linear approximation since $g_\rho \notin \mathcal{L}_\rho$, and in particular, for $r = 0$ we are making no assumption. Intuitively, for $0 < r < 1/2$, the condition quantifies *how far $g_\rho$ is from $\mathcal{L}_\rho$*, that is to be well approximated by a linear function.

[2]In a recent manuscript, it has been proved that this is indeed the minimax lower bound (G. Blanchard, personal communication)

[3] Available at http://www.cs.toronto.edu/~delve/data/comp-activ/desc.html

[4] Adult and Breast Cancer Wisconsin (Diagnostic), UCI repository, 2013.

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
