[Supplementary Material]

# A  Some Special Cases of Interest

Finally, before proving our main results, we illustrate and discuss a few special instances of the considered setting and related quantities.

**Linear and Functional Regression**  In classical linear regression, data are described by the following model

$$y_i = w_*^T x_i + \delta_i, \quad i = 1, \ldots, n$$

where $\delta_i$, $i = 1, \ldots, n$, are i.i.d. sample from a normal distribution and $w_*, x_1, \ldots, x_n \in \mathbb{R}^d$, $d \in \mathbb{N}^*$. In fixed design regression, the inputs $x_1, \ldots, x_n$ are assumed to be fixed, while in random design regression they are random sample according to some fixed unknown distribution [30]. It is easy to see that this latter setting is a special case of the framework in the paper (indeed the analysis in the paper can be also adapted with minor modifications to the fixed design setting). The regression model can be further complicated assuming the function of interest to be non linear (while we might still restrict the search of a solution to linear estimators). This can be dealt with for example considering kernel methods as we discuss below. Another special case of the setting in the paper is that of functional regression, where the input points are assumed to be infinite dimensional objects, for example curves, and they are formally described as functions in a Hilbert space. Clearly also this example is subsumed as a special case of our setting.

**Learning with Kernels**  The setting in the paper reduces to nonparametric learning in RKHS as a special case. Let $\Xi \times \mathbb{R}$ be a probability space with distribution $\mu$, that be can seen as the input/output space. The goal is then to minimize the risk, that, considering the square loss function, is given by

$$\mathcal{E}(f) = \int_{\Xi \times \mathbb{R}} (y - f(\xi))^2 d\mu(\xi, y) \tag{21}$$

and is well defined for all measurable functions. A common way to build an estimator is to consider a symmetric kernel $K : \Xi \times \Xi \to \mathbb{R}$ which is positive definite, that is for which the matrix with entries $K(\xi_i, \xi_j)$, $i, j = 1 \ldots n$, is positive semidefinite for all in $\xi_1, \ldots, \xi_n \in \Xi$, $n \in \mathbb{N}^*$. Such a kernel defines a unique Hilbert space of function $\mathcal{H}_K$ with inner product $\langle \cdot, \cdot \rangle_K$ and such that for all $\xi \in \Xi$, $K_\xi(\cdot) = K(\xi, \cdot) \in \mathcal{H}_K$ and the following reproducing property holds for all $f \in \mathcal{H}_K$, $f(\xi) = \langle f, K_\xi \rangle_K$. To see how this setting is subsumed by the one in the paper, it is useful to introduce the (feature) map $\Phi : \Xi \to \mathcal{H}_K$, where $\Phi(\xi) = K_\xi$, for $\xi \in \Xi$ and further consider $\overline{\Phi} : \Xi \times \mathbb{R} \to \mathcal{H}_K \times \mathbb{R}$, where $\overline{\Phi}(\xi, y) = (K_\xi, y)$, for $\xi \in \Xi$ and $y \in \mathbb{R}$. Assuming the kernel to be measurable, we can view $\overline{\Phi}$ as a random variable. If we denote its distribution on $\mathcal{H}_K \times \mathbb{R}$ by $\mu_{\overline{\Phi}}$, then we can let $\mathcal{H} = \mathcal{H}_K$ and $\rho = \mu_{\overline{\Phi}}$. It is known that the functions in a RKHS a measurable provided that the kernel is measurable [30], hence if we consider the risk of a function $f \in \mathcal{H}_K$ we have

$$\int_{\Xi \times \mathbb{R}} (y - f(\xi))^2 d\mu(\xi, y) = \int_{\Xi \times \mathbb{R}} (y - \langle f, K_\xi \rangle_K)^2 d\mu(\xi, y) = \int_{\mathcal{H} \times \mathbb{R}} (y - \langle f, x \rangle)^2 d\rho(x, y),$$

where we made the change of variables $(x, y) = (K_\xi, y) = \overline{\Phi}_\mu(\xi, y)$. As is well known in machine learning, we can view learning a function using a kernel as learning a linear function in suitable Hilbert space.

**Integral and Covariance Operators.**  The operator $L$ defined in (4) can be seen as an integral operator associated to a linear kernel and is closely related to the covariance operator, or rather the second moment operator defined by $\rho$. This connections allows to interpret Assumption 2 in terms of the principal components. To see this recall the definition of the linear operator

$$S : \mathcal{H} \to L^2(\mathcal{H}, \rho_X) : w \mapsto \langle w, \cdot \rangle_{\mathcal{H}},$$

introduced in Section 3.2.

Under Assumption 1 it is easy to see that $S$ is bounded, and its adjoint is given by

$$S^* : L^2(\mathcal{H}, \rho_X) \to \mathcal{H}, \quad S^* f = \int_{\mathcal{H}} x f(x) d\rho_X(x), \quad \forall f \in L^2(\mathcal{H}, \rho_X).$$

Then a straightforward calculation shows that $L = SS^*$. Moreover we can define $T : \mathcal{H} \to \mathcal{H}$ as $T = S^*S$ and check that

$$Tw = \int \langle x, w \rangle \, x d\rho_X(x), \quad \forall w \in \mathcal{H}.$$

The operator $T$ is the second moment operator associated to $\rho$ and its eigenfunctions are the principal components. Under Assumption 1, the operators $T, L$ are linear, positive, sef-adjoint and trace class, $S, S^*$ are bounded and Hilbert Schmidt, hence compact. The operators $T, L$ have the same non zero eigenvalues $(\sigma_j)_j$ which are the square of the singular values of $S$. If we denote by $(v_j)_j$ the eigenfunctions of $T$, the eigenfunctions of $L$ can be chosen to be $(u_j)_j$ with $u_j(x) = \sigma_j^{-1} \langle v_j, x \rangle_{\mathcal{H}}$, $\rho_X$-almost surely. This latter observation allows an interpretation of Condition (5). By considering higher fractional power we are essentially assuming that the regression function can be linearly approximated and its approximation can be effectively represented considering the principal components associated to large eigenvalues.

**Binary Classification** The results in the paper can be directly applied to binary classification. Indeed, in this setting the outputs are binary valued i.e. $\{-1, 1\}$ and the goal is to learn a classifier $c : \mathcal{H} \to \{-1, 1\}$ with small misclassification risk

$$R(c) = \mathbb{P}\left(c(X) \neq Y\right). \tag{22}$$

The above risk is minimized by the so called Bayes decision rule defined by $b_\rho(x) = \text{sign}(2\rho(1|x) - 1)$, $\rho_H$- almost surely, and where for $a \in \mathbb{R}$, $\text{sign}(a) = 1$, if $a \geq 1$ and $\text{sign}(a) = -1$ otherwise. A relaxation approach is usually considered to learn a classification rule, which is based on replacing the risk $R$ with a convex error functional defined over real valued functions, e.g. considering (21). A classification rule is then obtained by taking the sign.

So called comparison results quantify the cost of the relaxation. In particular, it is known that the following inequality relates $R$ and $\mathcal{E}$ defined in (22) and (21), respectively,

$$R(\text{sign}(f)) - R(b_\rho) \leq \sqrt{\mathcal{E}(f) - \mathcal{E}(f_\rho)}$$

for all measurable functions $f$. The latter inequality allows to derive excess misclassification risk and can be improved under additional assumption. We refer to [34] for further details in this direction.

# B  Proofs

In this appendix we prove the main results. The proof is quite long, and will be given relying on a series of lemmas.

## B.1  Preliminary Results

We collect very general results that will be applied to our setting.

Let $\mathcal{B}$ be a normed space. For every $r \in \mathbb{N}$, let $A_r : \mathcal{B} \to \mathcal{B}$ be a linear operator, let $(B_r)_{r \in \mathbb{N}}$ be a sequence in $\mathcal{B}$, and define the sequence $(X_r)_{r \in \mathbb{N}}$ in $\mathcal{B}$ recursively as

$$X_{r+1} = A_r X_r + B_r. \tag{23}$$

We repeatedly use the following well-known equality, which is valid for every $r \in \mathbb{N}^*$ and for every integer $s \leq r$,

$$X_r = \left(\prod_{i=s}^{r-1} A_i\right) X_s + \sum_{k=s}^{r-1} \left(\prod_{i=k+1}^{r-1} A_i\right) B_k. \tag{24}$$

We next state an auxiliary lemma, establishing the minimizing property of the gradient descent iteration also when the infimum is not attained. Despite the result is a basic property of a very classical algorithm, we were not able to find the proof of this fact. Convergence properties of gradient descent are usually studied in two settings: for differentiable functions (not necessarily convex) and for convex functions. In the first case, the typical results do not assume existence of a minimizer and establish convergence to zero of the gradient of the function [24]. In the convex setting, the minimizing property is established assuming the existence of a minimizer [24].

**Lemma 1.** *Let $\mathcal{H}$ be a Hilbert space, and $F\colon \mathcal{H} \to \mathbb{R}$ be a convex and differentiable function with $\beta$-Lipschitz continuous gradient. Let $v_0 \in \mathcal{H}$, let $(\eta)_{k\in\mathbb{N}}$ be such that, for every $k \in \mathbb{N}$, $\eta_k \in \,]0, 2/\beta[$ and define, for every $k \in \mathbb{N}$, $v_{k+1} = v_k - \eta_k \nabla F(v_k)$. Then*

$$(\forall u \in \mathcal{H}) \quad F(v_k) - F(u) \leq \frac{\|u - v_0\|_{\mathcal{H}}}{2\sum_{j=0}^{k} \eta_j} \tag{25}$$

*In particular, if $\sum_{k\in\mathbb{N}} \eta_k = +\infty$, $F(v_k) \to \inf F$.*

*Proof.* Since $F$ is convex and differentiable,

$$(\forall k \in \mathbb{N})(\forall u \in \mathcal{H}) \quad F(u) - F(v_k) \geq \langle \nabla F(v_k), u - v_k \rangle_{\mathcal{H}}$$
$$= \eta_k^{-1} \langle v_k - v_{k+1}, u - v_k \rangle_{\mathcal{H}}. \tag{26}$$

Therefore,

$$(\forall k \in \mathbb{N}) \quad 2\eta_k(F(u) - F(v_k)) \geq -2\langle v_{k+1} - v_k, u - v_k \rangle_{\mathcal{H}}$$
$$= \|u - v_{k+1}\|_{\mathcal{H}}^2 - \|v_{k+1} - v_k\|_{\mathcal{H}}^2 - \|u - v_k\|_{\mathcal{H}}^2$$
$$= \eta_k^2 \|\nabla F(v_k)\|_{\mathcal{H}}^2 + \|u - v_{k+1}\|_{\mathcal{H}}^2 - \|u - v_k\|_{\mathcal{H}}^2 \tag{27}$$

Let $t \in \mathbb{N}$ and define $\sigma_t = \sum_{k=0}^{t} \eta_k$. Summing (27) for $k = 0, \dots, t$ we obtain

$$2\sigma_t F(u) - 2\sum_{k=0}^{t} \eta_k F(v_k) \geq \sum_{k=0}^{t} \eta_k^2 \|\nabla F(v_k)\|_{\mathcal{H}} + \|u - v_{t+1}\|_{\mathcal{H}}^2 - \|u - v_0\|_{\mathcal{H}}^2. \tag{28}$$

Using the Lipschitz continuity of the gradient of $F$ (see [24, Equation (15) p.6]),

$$(\forall k \in \mathbb{N}) \quad F(v_k) - F(v_{k+1}) \geq \eta_k \left(1 - \frac{\eta_k \beta}{2}\right) \|\nabla F(v_k)\|_{\mathcal{H}}^2$$

Therefore,

$$(\forall k \in \mathbb{N}) \quad \sigma_k F(v_k) - \sigma_{k+1} F(v_{k+1}) + \eta_{k+1} F(v_{k+1}) \geq \sigma_k \eta_k \left(1 - \frac{\eta_k \beta}{2}\right) \|\nabla F(v_k)\|_{\mathcal{H}}^2 \tag{29}$$

Summing (29) for $k = 0, \dots, t-1$ we get, for every $t \in \mathbb{N}$

$$-\sigma_t F(v_t) + \sum_{k=0}^{t} \eta_k F(v_k) \geq \sum_{k=0}^{t-1} \sigma_k \eta_k \left(1 - \frac{\eta_k \beta}{2}\right) \|\nabla F(v_k)\|_{\mathcal{H}}^2 \tag{30}$$

Adding (30) to (28) we get, for every $u \in \mathcal{H}$

$$2\sigma_t(F(u) - F(v_t)) \geq \sum_{k=0}^{t} \eta_k^2 \|\nabla F(v_k)\|_{\mathcal{H}} + \|u - v_{t+1}\|_{\mathcal{H}}^2$$

$$- \|u - v_0\|_{\mathcal{H}}^2 + 2\sum_{k=0}^{t-1} \sigma_k \eta_k \left(1 - \frac{\eta_k \beta}{2}\right) \|\nabla F(v_k)\|_{\mathcal{H}}^2$$

and hence,

$$(\forall u \in \mathcal{H}) \quad F(v_t) - F(u) \leq \frac{\|u - v_0\|_{\mathcal{H}}^2}{2\sigma_t}.$$

$\square$

We next recall a probabilistic inequality for martingales [23, Theorem 3.4] (see also [32, Lemma A.1 and Corollaries A.2 and A.3]).

**Theorem 4.** *Let $(\xi_i, \mathcal{F}_i)_{1 \leq i \leq n}$ be an adapted family of random vectors taking values in a Hilbert space with norm $\|\cdot\|$, such that $\mathbb{E}[\xi_i | \mathcal{F}_{i-1}] = 0$ a.s. Assume that there exist $M \in \mathbb{R}_{++}$ such that $\|\xi_i\| \leq M$. Then, for every $\delta \in \,]0, 1[$ the following holds*

$$\mathbb{P}\left(\left\{\sup_{1 \leq j \leq n} \left\|\frac{1}{n}\sum_{i=1}^{j} \xi_i\right\| \leq \frac{8M}{3\sqrt{n}} \log \frac{2}{\delta}\right\}\right) \geq 1 - \delta.$$

## B.2 Proof of STEP 1

Here we first introduce a recursive expression which is satisfied by the sequence $(\hat{w}_t)_{t \in \mathbb{N}}$, that allows to interpret the incremental gradient iteration as a gradient descent iteration with errors. To do so, we start by introducing some further notation and then show that the iteration presented in (7)-(8) results from the application of the incremental gradient method to the empirical risk.

Consider the operators $S$ and $S_x \colon \mathcal{H} \to \mathbb{R}$ introduced in Section 3.2. Then $S_x$ is a bounded linear operator and $\|S_x\| \leq \|x\|_{\mathcal{H}}$. Using these linear operators, Problems (1) and (9) can be expressed as convex quadratic minimization problems. The empirical risk can be written as

$$\hat{\mathcal{E}}(w) = \frac{1}{n} \sum_{i=1}^{n} (S_{x_i} w - y_i)^2, \tag{31}$$

and recalling Assumption 2 and (19), we have

$$\mathcal{E}(w) = \int_{\mathcal{H} \times \mathbb{R}} (Sw(x) - y)^2 d\rho(x, y) = \|Sw - g_\rho\|_\rho^2 + \inf_{\mathcal{H}} \mathcal{E}. \tag{32}$$

Using the operators $T$ and $T_x$ introduced in Section refsec:dec, and computing the gradients of $\mathcal{E}$ and $\hat{\mathcal{E}}$ respectively, (7)-(8) can be rewritten as

$$\hat{u}_t^0 = \hat{w}_t; \qquad \hat{u}_t^i = \hat{u}_t^{i-1} - \frac{\gamma}{n}(T_{x_i} \hat{u}_t^{i-1} - S_{x_i}^* y_i), \quad i = 1, \ldots, n \tag{33}$$

and (17)-(18) can be expressed as

$$u_t^0 = w_t; \qquad u_t^i = u_t^{i-1} - \frac{\gamma}{n}(T u_t^{i-1} - S^* g_\rho). \tag{34}$$

It is apparent from (33) that the considered iteration is derived from the application of the incremental gradient algorithm to the empirical error (see [4, 19]). At the same time (34) shows that iteration (17)-(18) can be seen as the result of applying the incremental gradient descent algorithm to the expected loss, which clearly, for fixed $n$, can be written as

$$w \mapsto \frac{1}{n} \sum_{i=1}^{n} \mathcal{E}(w).$$

**Lemma 2.** *Let $t \in \mathbb{N}$, and let $w_t$ be defined as in (17)-(18), with $w_0 = 0$. Let $\eta = \gamma/n$. Then*

$$w_t = \eta \sum_{j=0}^{nt-1} \left(I - \eta T\right)^{nt-j-1} S^* g_\rho. \tag{35}$$

*Proof.* Let, for every $k \in \mathbb{N}$,
$$v_{k+1} = (I - \eta T)v_k + \eta S^* g_\rho.$$
Then, by (34), $w_t = v_{nt}$, and (35) follows. $\qquad\square$

In other words, the statement of Lemma 2 states that the $t$-th epoch of the incremental gradient descent iteration in (17)-(18) coincides with $n$ steps of gradient descent with stepsize $\gamma/n$.

Next, we relate the iteration (7)-(8) to the gradient descent iteration on the empirical error. These will be used in the error analysis and provide some useful comparison between these two methods. Hereafter, we will use the operator $\hat{T}$ intorduced in Section 3.2.

The following lemma provides an alternative expression for the composition of linear operators.

**Lemma 3.** *Let $n \in \mathbb{N}^*$, let $(T_i)_{1 \leq i \leq n}$ be a family of linear operators from $\mathcal{H}$ to $\mathcal{H}$, and let $(w_i)_{1 \leq i \leq n} \in \mathcal{H}^n$. Then*

$$\prod_{i=1}^{n}(I - T_i) = I - \sum_{j=1}^{n} T_j + \sum_{k=2}^{n} \left( \prod_{i=k+1}^{n} (I - T_i) \right) T_k \sum_{j=1}^{k-1} T_j \tag{36}$$

*and*

$$\sum_{i=1}^{n} \left( \prod_{k=i+1}^{n} (I - T_k) \right) w_i = \sum_{i=1}^{n} w_i - \sum_{k=2}^{n} \left( \prod_{i=k+1}^{n} (I - T_i) \right) T_k \sum_{j=1}^{k-1} w_j \tag{37}$$

*Proof.* By induction. Equality (36) is trivially satisfied for $n = 1$. Suppose now that $n \geq 2$, and that (36) holds for $n - 1$. Then

$$\prod_{i=1}^{n}(I - T_i) = (I - T_n)\prod_{i=1}^{n-1}(I - T_i)$$

$$= (I - T_n)\left(I - \sum_{j=1}^{n-1}T_j + \sum_{k=2}^{n-1}\left(\prod_{i=k+1}^{n-1}(I - T_i)\right)T_k\sum_{j=1}^{k-1}T_j\right)$$

$$= I - \sum_{j=1}^{n}T_j + T_n\sum_{j=1}^{n-1}T_j + \sum_{k=2}^{n-1}\left(\prod_{i=k+1}^{n}(I - T_i)\right)T_k\sum_{j=1}^{k-1}T_j$$

$$= I - \sum_{j=1}^{n}T_j + \sum_{k=2}^{n}\left(\prod_{i=k+1}^{n}(I - T_i)\right)T_k\sum_{j=1}^{k-1}T_j,$$

and the validty of (36) for every $n \in \mathbb{N}^*$ follows by induction.

Equality (37) is trivially satisfied for $n = 1$. Suppose now that $n \geq 2$ and that (36) holds for $n - 1$. Then

$$\sum_{i=1}^{n}\left(\prod_{k=i+1}^{n}(I - T_k)\right)w_i = \sum_{i=1}^{n-1}\left(\prod_{k=i+1}^{n}(I - T_k)\right)w_i + w_n$$

$$= (I - T_n)\left(\sum_{i=1}^{n-1}w_i - \sum_{k=2}^{n-1}\left(\prod_{i=k+1}^{n-1}(I - T_i)\right)T_k\sum_{j=1}^{k-1}w_j\right) + w_n$$

$$= \sum_{i=1}^{n}w_i - T_n\sum_{j=1}^{n-1}w_j - \sum_{k=2}^{n-1}\left(\prod_{i=k+1}^{n}(I - T_i)\right)T_k\sum_{j=1}^{k-1}w_j$$

$$= \sum_{i=1}^{n}w_i - \sum_{k=2}^{n}\left(\prod_{i=k+1}^{n}(I - T_i)\right)T_k\sum_{j=1}^{k-1}w_j,$$

and the conclusion follows. $\qquad\square$

The following lemma establishes an equivalent expression for the iterates (7)-(8).

**Lemma 4.** *Let $t \in \mathbb{N}$. Then,*

$$\hat{w}_{t+1} = \prod_{i=1}^{n}\left(I - \frac{\gamma}{n}T_{x_i}\right)\hat{w}_t + \frac{\gamma}{n}\sum_{i=1}^{n}\prod_{k=i+1}^{n}\left(I - \frac{\gamma}{n}T_{x_k}\right)S^*_{x_i}y_i \qquad (38)$$

*Proof.* For every $i \in \{1, \ldots, n\}$, the update of $\hat{v}^i_t$ in (8) can be equivalently written as in (33), and is of the form (23), with $A_r = I - (\gamma/n)T_{x_{r+1}}$, and $B_r = (\gamma/n)S^*_{x_{r+1}}y_{r+1}$, for every $r \in \{0, \ldots, n-1\}$, and $X_0 = \hat{w}_t$. Equation (38) follows by writing (24) for $r = n - 1$. $\qquad\square$

**Proposition 1.** *For every $t \in \mathbb{N}$, the iteration (7)-(8) can be written as*

$$\hat{w}_{t+1} = (I - \gamma\hat{T})\hat{w}_t + \gamma\left(\frac{1}{n}\sum_{j=1}^{n}S^*_{x_j}y_j\right) + \gamma^2\left(\hat{A}\hat{w}_t - \hat{b}\right), \qquad (39)$$

*with*

$$\hat{A} = \frac{1}{n^2}\sum_{k=2}^{n}\prod_{i=k+1}^{n}\left(I - \frac{\gamma}{n}T_{x_i}\right)T_{x_k}\sum_{j=1}^{k-1}T_{x_j}, \quad \hat{b} = \frac{1}{n^2}\sum_{k=2}^{n}\prod_{i=k+1}^{n}\left(I - \frac{\gamma}{n}T_{x_i}\right)T_{x_k}\sum_{j=1}^{k-1}S^*_{x_j}y_j. \qquad (40)$$

*The iteration (17)-(18) applied to $\mathcal{E}$ can be expressed as*

$$w_{t+1} = (I - \gamma T)w_t + \gamma S^* g_\rho + \gamma^2(Aw_t - b). \qquad (41)$$

*with*

$$A = \frac{1}{n^2} \sum_{k=2}^{n} \left[ \prod_{i=k+1}^{n} \left( I - \frac{\gamma}{n} T \right) \right] T \sum_{j=1}^{k-1} T, \quad b = \frac{1}{n^2} \sum_{k=2}^{n} \left[ \prod_{i=k+1}^{n} \left( I - \frac{\gamma}{n} T \right) \right] T \sum_{j=1}^{k-1} S^* g_\rho. \quad (42)$$

*Proof.* Equations (39) and (40) follow from Lemma 3 and Lemma 4 applied with $(\forall i \in \{1, \ldots, n\})$ $T_i = (\gamma/n) T_{x_i}$. Equations (41) and (42) follow from Lemma 3 and Lemma 4 applied with $(\forall i \in \{1, \ldots, n\})$ $T_i = (\gamma/n) T$. □

Note that, although not explicitly specified, the operator $A$ and the element $b \in \mathcal{H}$ in Proposition 1 depend on $n$. Equation (39) allows to compare the update resulting from one epoch of the iteration (7)-(8) with the one of a standard gradient descent on the empirical error with stepsize $\gamma$, which is given by

$$\hat{v}_{t+1} = \left( I - \frac{\gamma}{n} \sum_{j=1}^{n} T_{x_j} \right) \hat{v}_t + \frac{\gamma}{n} \sum_{j=1}^{n} S^*_{x_j} y_j \quad (43)$$

for an arbitrary $\hat{v}_0 \in \mathcal{H}$. As can be seen comparing (39) and (43), In particular, the incremental gradient descent can be interpreted as a perturbed gradient descent step, with perturbation

$$\hat{e}_t = \gamma^2 \left( \hat{A} \hat{w}_t - \hat{b} \right).$$

### B.3 Proof of STEP 2

The following recursive expression is key to get sample bounds estimates, and is at the basis of Lemma 7.

**Lemma 5.** *In the setting of Section 2, let $t \in \mathbb{N}$ and let $\hat{w}_t$ and $w_t$ be defined as in (7)-(8) and $w_t$ be defined as (17)-(18), respectively. Define $\hat{A}$ and $\hat{b}$ as in (40), and $A$ and $b$ as in (42). Then*

$$\hat{w}_t - w_t = \left( I - \gamma \hat{T} + \gamma^2 \hat{A} \right)^t (\hat{w}_0 - w_0) + \gamma \sum_{k=0}^{t-1} \left( I - \gamma \hat{T} + \gamma \hat{A} \right)^{t-k+1} \zeta_k \quad (44)$$

*with*

$$\zeta_k = (T - \hat{T}) w_k + \gamma (\hat{A} - A) w_k + \left( \frac{1}{n} \sum_{i=1}^{n} \hat{S}^*_{x_i} y_i - S^* g_\rho \right) + \gamma (b - \hat{b}). \quad (45)$$

*Proof.* We have

$$\hat{w}_{t+1} = (I - \gamma \hat{T} + \gamma^2 \hat{A}) \hat{w}_t + \frac{\gamma}{n} \sum_{i=1}^{n} \hat{S}^*_{x_i} y_i - \gamma^2 \hat{b}. \quad (46)$$

Adding and subtracting $(-\gamma \hat{T} + \gamma^2 \hat{A}) w_t$ it follows from (41) that

$$\hat{w}_{t+1} - w_{t+1} = (I - \gamma \hat{T} + \gamma^2 \hat{A})(\hat{w}_t - w_t) + \gamma (T - \hat{T}) w_t + \gamma^2 (\hat{A} - A) w_t$$
$$+ \gamma \left( \frac{1}{n} \sum_{i=1}^{n} \hat{S}^*_{x_i} y_i - S^* g_\rho \right) + \gamma^2 (b - \hat{b})$$

Relying on equation (23) we get (50). □

### B.4 Proof of STEP 3

Next we provide a lemma to bound the norm of the operator appearing in (50), and acting on the random variable $\zeta_k$ in (51).

**Lemma 6.** *In the setting of Section 2, let $\gamma \in \ ]0, n\kappa^{-1}]$. Then*

$$\| I - \gamma \hat{T} + \gamma^2 \hat{A} \| \le 1. \quad (47)$$

*Proof.* It follows from Lemma 3, equation (36) applied with $T_i = T_{x_i}$ and the definition of $\hat{A}_k$ in (40) that

$$I - \gamma\hat{T} + \gamma^2\hat{A} = \prod_{i=1}^{n}\left(I - \frac{\gamma}{n}T_{x_i}\right). \qquad (48)$$

Since $\|T_{x_i}\| \leq \kappa$ and by assumption $\gamma/n \leq \kappa^{-1}$, $\|I - (\gamma/n)T_{x_i}\| \leq 1$ and the statement follows. $\qquad \square$

We next provide a first inequality for the sample error.

**Lemma 7.** *Let $t \in \mathbb{N}$ and let $\hat{w}_t$ and $w_t$ be defined as in (7)-(8) and in (17)-(18), respectively. Define $\hat{A}$ and $\hat{b}$ as in (40), and $A$ and $b$ as in (42). Then*

$$\|\hat{w}_t - w_t\|_{\mathcal{H}} \leq \gamma\big(\|T-\hat{T}\|+\gamma\|\hat{A}-A\|\big)\sum_{k=0}^{t-1}\|w_k\|_{\mathcal{H}}+\gamma t\Big(\big\|\frac{1}{n}\sum_{i=1}^{n}\hat{S}_{x_i}^{*}y_i-S^{*}g_\rho\big\|+\gamma\|b-\hat{b}\|\Big). \quad (49)$$

*Proof.* By Lemma 5 we derive that

$$\hat{w}_t - w_t = \left(I - \gamma\hat{T} + \gamma^2\hat{A}\right)^{t}(\hat{w}_0 - w_0) + \gamma\sum_{k=0}^{t-1}\left(I - \gamma\hat{T} + \gamma\hat{A}\right)^{t-k+1}\zeta_k \qquad (50)$$

with

$$\zeta_k = (T - \hat{T})w_k + \gamma(\hat{A} - A)w_k + \left(\frac{1}{n}\sum_{i=1}^{n}\hat{S}_{x_i}^{*}y_i - S^{*}g_\rho\right) + \gamma(b - \hat{b}). \qquad (51)$$

Since $w_0 = \hat{w}_0$, we have

$$\|\hat{w}_t - w_t\|_{\mathcal{H}} \leq \gamma\sum_{k=0}^{t-1}\|I - \gamma\hat{T} + \gamma^2\hat{A}\|^{k+1}\|\zeta_k\| \qquad (52)$$

By Lemma 6, for every $j \in \mathbb{N}$, $\|I - \gamma\hat{T} + \gamma^2\hat{A}\| \leq 1$, and therefore

$$\|\hat{w}_t - w_t\|_{\mathcal{H}} \leq \gamma(\|T - \hat{T}\| + \gamma\|\hat{A} - A\|)\sum_{k=0}^{t-1}\|w_k\| + \left(\gamma\|\frac{1}{n}\sum_{i=1}^{n}\hat{S}_{x_i}^{*}y_i - S^{*}g_\rho\| + \gamma^2\|b - \hat{b}\|\right)t$$

$\qquad \square$

## B.5 Proof of STEP 4

Here, we provide a bound for $\|w_t\|_{\mathcal{H}}$. The proof technique is similar to that of known results in inverse problems [15], although the bound obtained in (53) is novel.

**Lemma 8.** *In the setting of Section 2, let Assumption 1 hold, let $n \in \mathbb{N}^*$, and let $\gamma \in\ ]0, n\kappa^{-1}[$. Let $w_0 = 0$, and let $t \in \mathbb{N}$. Then the following hold:*

(i) *Let Assumption 2 hold with $r \in [0, 1/2]$. Then*

$$\|w_t\|_{\mathcal{H}} \leq \max\{\kappa^{r-1/2}, (\gamma t)^{1/2-r}\}\|g\|_\rho . \qquad (53)$$

(ii) *Suppose that $\mathcal{O}$ is nonempty. Then there exist $w^\dagger$ as in (3) and $\beta \in\ ]0, +\infty[$ such that*

$$\|w_{t+1}\|_{\mathcal{H}} \leq \beta.$$

(iii) *Let Assumption 2 hold with $r \in\ ]1/2, +\infty[$. Then $w^\dagger$ is well defined and*

$$\|w_{t+1}\|_{\mathcal{H}} \leq \kappa^{r-1/2}\|g\|_\rho . \qquad (54)$$

*Proof.* Let $\epsilon \in \,]0, \kappa]$ and set $\eta = \gamma/n$. By Lemma 2 and by the spectral theorem [15, equation (2.43)], we derive

$$w_{t+1} = \sum_{j=0}^{nt-1} S^* \eta \prod_{i=j+1}^{nt-1} (I - \eta L) g_\rho .$$ (55)

(i): By (55)

$$\|w_{t+1}\|_{\mathcal{H}} \leq \eta \left\| S^* L^r \sum_{j=0}^{nt-1} (I - \eta L)^{nt-j+1} \right\| \|g\|_\rho$$ (56)

$$\leq \sup_{\sigma \in [0, \kappa]} \sigma^{1/2+r} \left| \sum_{j=0}^{nt-1} \eta (1 - \eta\sigma)^{nt-j+1} \right| \|g\|_\rho$$

$$= \max \left\{ \sup_{\sigma \in [0, \epsilon]} nt\eta\sigma^{1/2+r}, \sup_{\sigma \in [\epsilon, \kappa]} \sigma^{r-1/2} \left( 1 - (1 - \eta\sigma)^{nt} \right) \right\} \|g\|_\rho$$

$$\leq \psi(\epsilon) \|g\|_\rho$$

where $(\forall \epsilon \in \mathbb{R}_+) \; \psi(\epsilon) = \max \left\{ \epsilon^{1/2+r} nt\eta, \epsilon^{r-1/2} \right\}$. Since $\epsilon$ is arbitrary,

$$\|w_{t+1}\|_{\mathcal{H}} \leq \inf_{\epsilon \in ]0, \kappa]} \psi(\epsilon) \|g\|_\rho .$$ (57)

Now note that $\epsilon \in \,]0, +\infty] \mapsto nt\eta\epsilon^{1/2+r}$ is strictly increasing, and has limit equal to zero at zero. On the contrary, $\epsilon \in \,]0, +\infty] \mapsto \epsilon^{r-1/2}$ is strictly decreasing and $\lim_{\epsilon \to 0^+} \epsilon^{r-1/2} = +\infty$. Hence, there exists a unique point $\bar{\epsilon} \in \,]0, +\infty]$ such that

$$nt\eta\bar{\epsilon}^{1/2+r} = \bar{\epsilon}^{r-1/2} \quad \text{and} \quad \psi(\epsilon) = \begin{cases} \epsilon^{r-1/2} & \text{if } \epsilon \in \,]0, \bar{\epsilon}] \\ nt\eta\epsilon^{1/2+r} & \text{if } \epsilon \in [\bar{\epsilon}, +\infty] \,, \end{cases}$$ (58)

therefore $\bar{\epsilon}$ is the unique minimizer of $\psi$. Solving for $\bar{\epsilon}$ in (58), we get $\bar{\epsilon} = (nt\eta)^{-1}$. We derive, again from (58), that

$$\min_{\epsilon \in ]0, \kappa]} \psi(\epsilon) = \max\{\kappa^{r-1/2}, (\gamma t)^{1/2-r}\}.$$

Finally, (57) yields

$$\|w_{t+1}\|_{\mathcal{H}} \leq \left( \gamma t \right)^{1/2-r} \|g\|_\rho .$$

(ii): First note that, by Fermat's rule, $S^* g_\rho = T w^\dagger$. It follows form Lemma 2 that

$$w_t - w^\dagger = \left( \sum_{j=0}^{nt-1} \eta T (I - \eta T)^{nt-j+1} - I \right) w^\dagger$$

$$= (I - \eta T)^{nt} w^\dagger.$$

Let $(\sigma_m, h_m)_{m \in \mathbb{N}}$ be an eigensystem of $T$. Since $w^\dagger \in N(T)^\perp$ (see [15, Proposition 2.3]), it follows that $w^\dagger = \sum_{m \in \mathbb{N}} \langle w^\dagger, h_m \rangle h_m$. Therefore,

$$\|w_t - w^\dagger\|_{\mathcal{H}}^2 = \sum_{m \in \mathbb{N}} \left| (1 - \eta\sigma_m)^{nt} \langle w^\dagger, h_m \rangle \right|^2$$ (59)

Since, for every $m \in \mathbb{N}$, each summand is bounded by $|\langle w^\dagger, h_m \rangle|^2$, and $\sum_{m \in \mathbb{N}} |\langle w^\dagger, h_m \rangle|^2$, the Dominated Convergence Theorem yields

$$\lim_{t \to +\infty} \|w_t - w^\dagger\|_{\mathcal{H}}^2 = \sum_{m \in \mathbb{N}} \lim_{t \to +\infty} \left| (1 - \eta\sigma_m)^{nt} \langle w^\dagger, h_m \rangle \right|^2 = 0.$$ (60)

Hence, the sequence $(\|w_t\|_{\mathcal{H}})_{t \in \mathbb{N}}$ is bounded.

(iii): Arguing as in the proof of (i), it follows from (55) and Assumption 2 that

$$\|w_{t+1}\|_{\mathcal{H}} \leq \sup_{\sigma \in [0, \kappa]} \sigma^{r-1/2} \left( 1 - (1 - \eta\sigma)^{nt} \right) \right\} \|g\|_\rho$$

$$\leq \kappa^{r-1/2} \|g\|_\rho$$

$\square$

## B.6 Proof of STEP 5

The main novel probabilistic estimates are given in the following proposition. This is the more involved part of the proof, where many tricks are needed in order to get a manageable expression. The proof is based on writing the terms $\hat{A} - A$ and $\hat{b} - b$ as a sum of martingales and then apply Theorem 4 to derive concentration inequalities. We start with the following well-known lemma, which is a direct consequence of Theorem 4 (see also [13]).

**Lemma 9.** *In the Setting of Section 2, let Assumption 1 hold. For every $\delta \in\ ]0, 1]$*

$$\mathbb{P}\left(\left\|\frac{1}{n}\sum_{i=1}^{n}T_{x_i} - T\right\|_{HS} \le \frac{16\kappa}{3\sqrt{n}}\log\frac{2}{\delta}\right) \ge 1 - \delta, \tag{61}$$

*and*

$$\mathbb{P}\left(\left\|\frac{1}{n}\sum_{i=1}^{n}S_{x_i}^{*}y_i - S^{*}g_\rho\right\|_{\mathcal{H}} \le \frac{16\sqrt{\kappa}M}{3\sqrt{n}}\log\frac{2}{\delta}\right) \ge 1 - \delta. \tag{62}$$

*Proof.* Equation 61 follows from Theorem 4, since $(T_{x_i} - T)_{1 \le i \le n}$ is a family of i.i.d. random operators taking values in the space of Hilbert-Schmidt operators satisfying $\|T\|_{HS} \le \kappa$ and $\|T_{x_i}\|_{HS} \le \kappa$ (see also [13]). Equation 62 follows from Theorem 4 applied to the i.i.d. random vectors $(S_{x_i}^{*}y_i - S^{*}f_\rho)_{1 \le i \le n}$ in $\mathcal{H}$ whose norms are bounded by $2\kappa M$. $\qquad\square$

**Proposition 2.** *In the Setting of Section 2, let Assumption 1 hold, let $\gamma \in\ ]0, n\kappa^{-1}[$, and let $\delta \in\ ]0, 1]$. Then*

$$\mathbb{P}\left(\|\hat{A} - A\|_{HS} \le \frac{32\kappa^2}{3\sqrt{n}}\log\frac{4}{\delta}\right) \ge 1 - \delta, \tag{63}$$

*and*

$$\mathbb{P}\left(\|\hat{b} - b\|_{\mathcal{H}} \le \frac{32\kappa M^2}{3\sqrt{n}}\log\frac{4}{\delta}\right) \ge 1 - \delta. \tag{64}$$

*Proof.* We first show a useful decomposition. Recall that

$$\hat{A} = \frac{1}{n}\sum_{j=2}^{n}\frac{1}{n}\prod_{i=j+1}^{n}\left(I - \gamma T_{x_i}\right)T_{x_j}\sum_{l=1}^{j-1}T_{x_l} \qquad A = \frac{1}{n}\sum_{j=2}^{n}\frac{1}{n}\prod_{i=j+1}^{n}\left(I - \gamma T\right)T\sum_{l=1}^{j-1}T.$$

For every $j \in \{2, \ldots, n\}$, set

$$\hat{B}_j = \left[\prod_{i=j+1}^{n}\left(I - \gamma T_{x_i}\right)\right]T_{x_j}, \qquad B_j = \left[\prod_{i=j+1}^{n}\left(I - \gamma T\right)\right]T$$

we have

$$\hat{A} - A = \frac{1}{n}\sum_{j=2}^{n}\hat{B}_j\left(\frac{1}{n}\sum_{l=1}^{j-1}T_{x_l}\right) - \frac{1}{n}\sum_{j=2}^{n}B_j\frac{j-1}{n}T$$

$$= \frac{1}{n}\left[\sum_{j=2}^{n}\hat{B}_j\left(\frac{1}{n}\sum_{l=1}^{j-1}\left(T_{x_l} - T\right)\right) + Q_n T\right], \tag{65}$$

with

$$Q_n = \sum_{j=2}^{n}(\hat{B}_j - B_j). \tag{66}$$

We next bound each term appearing in (65). By Lemma 9, with probability greater than $1 - \delta$,

$$\sup_{j \in \{1, \ldots, n\}}\left\|\frac{1}{n}\sum_{l=1}^{j-1}\left(T_{x_l} - T\right)\right\|_{HS} \le \frac{16\kappa}{3\sqrt{n}}\log\frac{2}{\delta}. \tag{67}$$

On the other hand

$$\|\hat{B}_j\| \leq \prod_{i=j+1}^{n} \left\| I - \frac{\gamma}{n} T_{x_i} \right\| \|T_{x_j}\| \leq \kappa. \tag{68}$$

Note that $\sum_{j=2}^{n} \hat{B}_j \left( 1/n \sum_{l=1}^{j-1} (T_{x_l} - T) \right)$ is Hilbert-Schmidt, for $T_{x_l}$ and $T$ are Hilbert-Schmidt operators, with $\|T_{x_l}\|_{HS} \leq \kappa$ and $\|T\|_{HS} \leq \kappa$, and the family of Hilbert-Schmidt operators is an ideal with respect to the composition in $L(\mathcal{H})$. Therefore, by (67) and (68),

$$\frac{1}{n} \left\| \sum_{j=2}^{n} \hat{B}_j \left( \frac{1}{n} \sum_{l=1}^{j-1} (T_{x_l} - T) \right) \right\|_{HS} \leq \frac{16\kappa^2}{3\sqrt{n}} \log \frac{2}{\delta} \tag{69}$$

holds with probability greater than $1 - \delta$, for any $\delta \in ]0, 1[$. Next we write the quantity $Q_n$ appearing in the second term in (65) as the sum of a martingale. For short, we set $\eta = \frac{\gamma}{n}$ and for all $j \in \{2, \ldots, n\}$ we denote

$$\hat{\Pi}_j^n = \prod_{i=j+1}^{n} (I - \eta T_{x_i}), \qquad \Pi_j^n = \prod_{i=j+1}^{n} (I - \eta T),$$

so that from the definition of $Q_n$ in (66),

$$Q_n = \sum_{j=2}^{n} (\hat{\Pi}_j^n T_{x_j} - \Pi_j^n T).$$

We can derive a recursive update that determine a different expression for the quantity $Q_n$ as follows. Let $s \in \{1, \ldots, n-1\}$.

$$Q_{s+1} = \sum_{j=2}^{s+1} (\hat{\Pi}_j^{s+1} T_{x_j} - \Pi_j^{s+1} T)$$

$$= (T_{x_{s+1}} - T) + \sum_{j=2}^{s} (\hat{\Pi}_j^{s+1} T_{x_j} - \Pi_j^{s+1} T)$$

$$= (T_{x_{s+1}} - T) + \sum_{j=2}^{s} ((I - \eta T_{x_{s+1}}) \hat{\Pi}_j^s T_{x_j} - (I - \eta T) \Pi_j^s T)$$

$$= (T_{x_{s+1}} - T) + (I - \eta T_{x_{s+1}}) \sum_{j=2}^{s} (\hat{\Pi}_j^s T_{x_j} - \Pi_j^s T) + \eta (T - T_{x_{s+1}}) \sum_{j=2}^{s} \Pi_j^s T$$

$$= (I - \eta T_{x_{s+1}}) Q_s + (T_{x_{s+1}} - T) \left( I - \eta \sum_{j=2}^{n} \Pi_j^s T \right).$$

Applying equation (23), since $Q_1 = 0$, we get

$$Q_n = \sum_{l=1}^{n} \Theta_l \tag{70}$$

where, $\Theta_1 = 0$ and, for every $l \in \{2, \ldots, n\}$,

$$\Theta_l = \prod_{i=l+1}^{n} \left( I - \frac{\gamma}{n} T_{x_i} \right) (T_{x_l} - T) \left( I - \frac{\gamma}{n} \sum_{j=2}^{l-1} \prod_{i=j+1}^{l-1} \left( I - \frac{\gamma}{n} T \right) T \right).$$

For every $l = 1, \ldots, n$

$$\mathbb{E}[\Theta_l] = 0,$$

being $T_{x_2}, \ldots, T_{x_n}$ independent and $\mathbb{E}[(T_{x_l} - T)] = 0$. Moreover the conditional expectation

$$\mathbb{E}[\Theta_l \mid \Theta_{l+1}, \ldots, \Theta_n] = 0,$$

since $T_{x_l}$ is independent from $T_{x_{l+1}}, \ldots, T_{x_n}$. Therefore the sequence $(\Theta_l)_{1 \le l \le n}$ is a martingale difference sequence. The operator $\Theta_l$ is Hilbert-Schmidt, since it is the composition of a Hilbert-Schmidt operator with a continuous one. Moreover, $\|\Theta_1\| = 0$. Next, since the operator $T$ is compact and self-adjoint and $0 \le \gamma/n \le 1/\|T\|$, from the spectral mapping theorem, for every $l \in \{2, \ldots, n\}$

$$\left\| I - \frac{\gamma}{n} \sum_{j=2}^{l-1} \prod_{i=j+1}^{l-1} \left( I - \frac{\gamma}{n} T \right) T \right\| = \sup_{x \in [0,1]} \left| 1 - \sum_{j=2}^{l-1} x \left( 1 - x \right)^{l-j-1} \right|.$$

We have

$$0 \le \sum_{j=2}^{l-1} x \left( 1 - x \right)^{l-j-1} = \sum_{j=2}^{l-1} \left( (1-x)^{l-j-1} - (1-x)^{l-j} \right) = 1 - (1-x)^{l-2} \le 1$$

Therefore

$$\left\| I - \frac{\gamma}{n} \sum_{j=2}^{l-1} \prod_{i=j+1}^{l-1} \left( I - \frac{\gamma}{n} T \right) T \right\| \le 1 \, .$$

Using the last inequality, we derive

$$\|\Theta_l\|_{HS} \le \left\| \prod_{i=l+1}^{n} \left( I - \frac{\gamma}{n} T_{x_i} \right) \right\| \left\| T_{x_l} - T \right\|_{HS} \left\| I - \frac{\gamma}{n} \sum_{j=2}^{l-1} \prod_{i=j+1}^{l-1} \left( I - \frac{\gamma}{n} T \right) T \right\|$$

$$\le \|T_{x_l} - T\|_{HS}$$

$$\le 2\kappa \, .$$

Then, Theorem 4 applied to $(\Theta_l)_{i \le l \le n}$, yields

$$\left\| \frac{1}{n} \sum_{l=1}^{n} \Theta_l \right\|_{HS} \le \frac{16\kappa}{3\sqrt{n}} \log \frac{2}{\delta} \tag{71}$$

with probability greater than $1 - \delta$. Therefore, with probability greater than $1 - \delta$

$$\frac{1}{n} \left\| Q_n T \right\|_{HS} \le \frac{16\kappa^2}{3\sqrt{n}} \log \frac{2}{\delta}. \tag{72}$$

The statement then follows recalling the decomposition in (65), and summing (72) with (69). From the definition of $\hat{b}$ and $b$ in equations (39) and (42) respectively, we have

$$\hat{b} - b = \frac{1}{n^2} \sum_{j=2}^{n} \hat{B}_j \sum_{l=1}^{j-1} S_{x_l}^* y_l - \frac{1}{n^2} \sum_{j=2}^{n} B_j \sum_{l=1}^{j-1} S^* g_\rho \, , \tag{73}$$

and equation (64) follows reasoning as in the previous part of the proof. $\qquad \square$

## B.7 Sample error

The proof of the bound on the sample error easily follows from the above results.

**Theorem 5** (Sample error). *Let Assumption 1 hold. Let $n \in \mathbb{N}^*$, suppose that $\gamma \in \,]0, n\kappa^{-1}]$, and let $\hat{w}_0 = w_0 = 0$. Let $\delta \in \,]0, 1[$, and, for every $t \in \mathbb{N}^*$, let $\hat{w}_t$ and $w_t$ be defined as in (7)-(8) and (17)-(18), respectively. Then the following hold:*

(i) *Let Assumption 2 hold, for some $r \in [0, 1/2]$, and let $t \in \mathbb{N}^*$. Then, with probability greater than $1 - \delta$*

$$\|\hat{w}_t - w_t\|_{\mathcal{H}} \le \frac{\log(16/\delta)}{3\sqrt{n}} \left[ (16\sqrt{\kappa}M + 32\kappa M^2 \gamma) \gamma t \right. \tag{74}$$

$$\left. + (16\kappa + 32\kappa^2 \gamma) \|g\|_\rho \max \left\{ \kappa^{r-1/2} \gamma t, \left( \frac{1 - 2r}{3 - 2r} + \frac{2}{3 - 2r} t^{3/2-r} \right) \gamma^{3/2-r} \right\} \right].$$

(ii) *Let $t \in \mathbb{N}^*$, and let Assumption 2 hold for some $r \in [1/2, +\infty]$. Then, with probability greater than $1 - \delta$*

$$\|\hat{w}_t - w_t\|_{\mathcal{H}} \leq \frac{\log(16/\delta)}{3\sqrt{n}} \left[ 16\sqrt{\kappa}M + 32\kappa M^2\gamma + (16\kappa + 32\kappa^2\gamma)\|g\|_{\rho}\kappa^{r-1/2} \right] \gamma t. \tag{75}$$

*Proof.* Substituting the bounds obtained in Lemma 9 and Proposition 2 with $\delta/4$ into (49), and applying Lemma 7, we obtain yield that with probability bigger than $1 - \delta$

$$\|\hat{w}_t - w_t\|_{\mathcal{H}} \leq \frac{\log(16/\delta)}{3\sqrt{n}} \left( \gamma(16\kappa + 32\kappa^2\gamma) \sum_{k=0}^{t-1} \|w_k\|_{\mathcal{H}} + \gamma t\left(16\sqrt{\kappa}M + 32\kappa M^2\gamma\right) \right).$$

Statements (i) and (ii) directly follow from the bound on $\|w_k\|_{\mathcal{H}}$ obtained in Lemma 8. □

### B.8 Proof of STEP 6 – approximation error

The proof of this result is similar to that of the approximation error bounds obtained in [34, 35], and uses spectral techniques, which are classical in linear inverse problems [15].

**Theorem 6** (Approximation error). *In the setting of Section 2, let Assumption 1 hold, let $n \in \mathbb{N}^*$, let $w_0 \in \mathcal{H}$, let $\gamma \in \,]0, n\kappa^{-1}[$ and let $(w_t)_{t\in\mathbb{N}}$ be defined as in (17)-(18). Then the following hold:*

(i) *The approximation error $\mathcal{E}(w_t) - \inf_{\mathcal{H}} \mathcal{E} \to 0$.*

(ii) *Suppose that $\mathcal{O}$ is nonempty. Then $w^{\dagger}$ in (3) exists and $\|w_t - w^{\dagger}\|_{\mathcal{H}} \to 0$.*

(iii) *Let Assumption 2 hold for some $r \in \,]0, +\infty[$. Then*

$$\mathcal{E}(w_t) - \inf_{\mathcal{H}} \mathcal{E} \leq \left( \frac{r}{\gamma t} \right)^{2r} \|g\|_{\rho}^2.$$

(iv) *Let Assumption 2 hold for some $r \in \,]1/2, +\infty[$. Then $w^{\dagger}$ in (3) is well-defined and*

$$\|w_t - w^{\dagger}\|_{\mathcal{H}} \leq \left( \frac{r - 1/2}{\gamma t} \right)^{r-1/2} \|g\|_{\rho}.$$

*Proof.* (i): This is a direct consequence of Lemma 1.

(ii): The proof is the same lines as that of Lemma 8-(ii). See in particular (60).

(iii): It follows from Lemma 2 that

$$\|Sv_{t+1} - g_{\rho}\|_{\rho} = \left\| \left( \left( \sum_{j=0}^{nt-1} L\eta(I - \eta L)^{nt-j+1} \right) - I \right) g_{\rho} \right\|_{\rho}$$

$$= \sup_{\sigma \in [0, \|L\|]} \sigma^r (1 - \eta\sigma)^{nt} \|g\|_{\rho}.$$

Note that, the last term is maximized at $\sigma = rn/(\gamma(r + nt + 1))$, hence for every $\sigma \in [0, +\infty[$,

$$\sigma^r \left( 1 - \frac{\gamma}{n}\sigma \right)^{nt} \leq \left( \frac{nr}{\gamma(r + nt)} \right)^r \left( 1 - \frac{r}{r + nt} \right)^{nt}$$

$$\leq \left( \frac{nr}{\gamma(r + nt)} \right)^r$$

$$\leq \left( \frac{r}{t\gamma} \right)^r.$$

Finally, the equality

$$\mathcal{E}(w_t) - \inf_{\mathcal{H}} \mathcal{E} = \|Sw_t - g_{\rho}\|_{\rho}^2,$$

yields the statement.

(iv): Since $L^{1/2}$ is a partial isometry between $L^2(\mathcal{H}, \rho_X)$ and $S(\mathcal{H})$ and $g_\rho = L^{1/2}(L^{r-1/2}g)$ by Assumption (2), it follows that $g_\rho \in S(\mathcal{H})$, and thus $\mathcal{O}$ is nonempty, and $w^\dagger$ is well defined. Moreover, since $S^* g_\rho = T w^\dagger$, we also get that $T w^\dagger = S^* L^r g = T^r S^* g$, implying that $w^\dagger = T^\dagger T^r S^* g$. It follows from Lemma 2 and [15, Equation 2.24] that

$$\|w_t - w^\dagger\|_{\mathcal{H}} \leq \left\| T^\dagger T^r \left( \sum_{j=0}^{nt-1} \eta T (I - \eta T)^{nt-j+1} - I \right) S^* \right\| \|g\|_\rho$$

$$= \sup_{\sigma \in [0, \|L\|]} \sigma^{r-1/2} (1 - \eta\sigma)^{nt} \|g\|_\rho$$

$$\leq \left( \frac{r - 1/2}{t\gamma} \right)^{r-1/2} \|g\|_\rho,$$

where the last inequality can be derived proceeding as in (iii). $\qquad\square$

### B.9   STEP 7: proof of the main results

Let us denote by $\mathcal{S}(t, n, \delta)$ the right hand side of (74). Then, combining the sample error estimate with the error decomposition (19), we can immediately derive the following inequalities

$$\mathcal{E}(\hat{w}_t) - \inf_{\mathcal{H}} \mathcal{E} \leq 2\kappa (\mathcal{S}(t, n, \delta))^2 + 2\mathcal{A}(t, \gamma, n)$$

$$\|\hat{w}_t - w^\dagger\|_{\mathcal{H}} \leq \mathcal{S}(t, n, \delta) + \|w_t - w^\dagger\|$$

with probability greater than $1 - \delta$. Note that analogous inequalities hold for the case $r > 1/2$ and with respect to the norm in $\mathcal{H}$. We are now ready to prove the Theorems stated in Section 3. The proof of universal consistency is a consequence of the sample and approximation error bounds, and of the application of Borel-Cantelli Lemma.

**Proof of Theorem 1**. (i): Recalling the error decomposition in (19), we have

$$\mathcal{E}(\hat{w}_t) - \inf_{\mathcal{H}} \mathcal{E} \leq 2\kappa \|\hat{w}_t - w_t\|_{\mathcal{H}}^2 + 2(\mathcal{E}(w_t) - \inf_{\mathcal{H}} \mathcal{E}).$$

Suppose that $t > 1$. Theorem 5 applied with $r = 0$ yields that there exists $c \in \mathbb{R}_{++}$ such that, with probability greater than $1 - \delta$

$$\|\hat{w}_t - w_t\|_{\mathcal{H}} \leq c \frac{t^{3/2} \log(16/\delta)}{3\sqrt{n}}. \tag{76}$$

Since $\sum_{k \in \mathbb{N}} \gamma = +\infty$ and $\gamma \leq \kappa^{-1} \leq n\kappa^{-1}$, by Theorem 6(i), $\mathcal{A}(t) = \mathcal{E}(w_t) - \inf_{\mathcal{H}} \mathcal{E} \to 0$. Moreover, $\mathcal{A}(t^*(n)) \to 0$ since $t^*(n) \to +\infty$. Let $\eta \in ]0, +\infty[$ and let $\alpha \in ]1, +\infty[$. By (10), there exists $\bar{n} \in \mathbb{N}$ such that, for every $n \geq \bar{n}$, $\eta n / t^*(n)^3 > \alpha \log n$. Define

$$A_{n,\eta} = \left\{ \mathcal{E}(\hat{w}_{t^*(n)}) - \inf_{\mathcal{H}} \mathcal{E} \geq \mathcal{A}(t^*(n)) + c\eta \right\}. \tag{77}$$

By (76), for every $n \geq \bar{n}$, $\mathbb{P}(A_{n,\eta}) \leq 16 \exp(-\eta n / t^*(n)^{3(1-\theta)}) \leq \exp(-\alpha \log n)$. Therefore,

$$\sum_{n \geq \bar{n}} \mathbb{P}(A_{n,\eta}) \leq \sum_{n \geq \bar{n}} n^{-\alpha} < +\infty,$$

hence the Borel-Cantelli lemma yields $\mathbb{P}(\bigcap_{k \geq \bar{n}} \bigcup_{n \geq k} A_{n,\eta}) = 0$, and almost sure convergence follows.

(ii): Since $\mathcal{O}$ is nonempty, it follows that there $g_\rho \in S(\mathcal{H}) = L^{1/2}(\mathcal{H})$. Therefore, Assumption 2 is satisfied with $r = 1/2$. From Theorem 5(ii), that there exists $c_1 \in ]0, +\infty[$ such that, with probability greater than $1 - \delta$

$$\|\hat{w}_t - w_t\|_{\mathcal{H}} \leq \frac{c_1 t \log(16/\delta)}{3\sqrt{n}} \tag{78}$$

Moreover, since $\sum_{k \in \mathbb{N}} \gamma = +\infty$ and $\gamma \leq n\kappa^{-1}$, by Theorem 6(ii), $\|w_t - w^\dagger\|_{\mathcal{H}} \to 0$. Reasoning as in (i), we obtain (12).

**Proof of Theorem 2**. (i): It follows from Theorem 5(ii), that with probability greater than $1 - \delta$

$$\|\hat{w}_t - w_t\|_{\mathcal{H}} \le \frac{32\log(16/\delta)}{n}\left[M\kappa^{-1/2} + 2M^2\kappa^{-1} + 3\kappa^{r-1/2}\|g\|_\rho\right]t \tag{79}$$

Moreover, Theorem 6(iv) yields

$$\|w_t - w^\dagger\|_{\mathcal{H}} \le \left(\frac{r-1/2}{\gamma t}\right)^{r-1/2}\|g\|_\rho. \tag{80}$$

Inequality (13) follows by adding (79) with (80).

(ii): Let $\alpha \in\, ]0, +\infty[$ and let $t^*(n) = \lceil n^\alpha \rceil$. Minimizing the right hand side in (13), we get

$$\alpha - 1/2 = \alpha(1/2 - r)$$

leading to the expression of $t^*(n)$. Now, let $n \in \mathbb{N}^*$ and $\beta \in [1, 2[$ be such that $n^\alpha \le t^*(n) = \beta n^\alpha \le n^\alpha + 1$. Then, by (13) we get

$$\|\hat{w}_t - w_t\|_{\mathcal{H}} \le \beta^{1/2-r}2r + 132\log(16/\delta)\left[M\kappa^{-1/2} + 2M^2\kappa^{-1} + 3\kappa^{r-1/2}\|g\|_\rho\right]n^{\frac{1/2-r}{2r+1}} \tag{81}$$

and

$$\|w_t - w^\dagger\|_{\mathcal{H}} \le \left(\frac{r-1/2}{\beta}\right)^{r-1/2}\|g\|_\rho n^{\frac{1/2-r}{2r+1}}. \tag{82}$$

Equation (14) follows recalling that $\beta \in [1, 2[$ in (i).

**Proof of Theorem 3**. (i): Recalling (19), it follows from Theorem 5(i) and Theorem 6(iii), that

$$\mathcal{E}(\hat{w}_t) - \inf_{\mathcal{H}}\mathcal{E} \le 2\frac{\left(32\log(16/\delta)\right)^2}{n}\left[M + 2M^2\kappa^{-1/2} + 3\kappa^r\|g\|_\rho\right]^2 t^2 + \left(\frac{r}{\gamma t}\right)^{2r}\|g\|_\rho^2 \tag{83}$$

(ii): As in the proof of Theorem 2(ii), set $t = \lceil n^\alpha \rceil$. Then, minimizing the right hand side in (83), we derive

$$2\alpha - 1 = -2\alpha r$$

which gives $\alpha = 1/(2r + 2)$. Let $n \in \mathbb{N}^*$ and $\beta \in [1, 2[$ be such that $n^\alpha \le t^*(n) = \beta n^\alpha \le n^\alpha + 1$. Then, plugging the expression of $t^*(n)$ into (83). we get

$$\mathcal{E}(\hat{w}_{t^*(n)}) - \inf_{\mathcal{H}}\mathcal{E} \le 8\frac{\left(32\log(16/\delta)\right)^2}{n}\left[M + 2M^2\kappa^{-1/2} + 3\kappa^r\,\|g\|_\rho\right]^2 n^{1/(r+1)}$$

$$+ 2\left(\frac{r}{\gamma}\right)^{2r}n^{-r/(r+1)}\|g\|_\rho^2.$$

# A    Non attainable case

**Theorem 7** (Finite sample bounds for the risk – non attainable case)**.** *In the setting of Section 2, let Assumption 1 hold, and let $\gamma \in\, ]0, \kappa^{-1}]$. Let Assumption 2 be satisfied for some $r \in\, ]0, 1/2]$. Then the following hold:*

(i) *For every $t \in \mathbb{N}^*$, with probability greater than $1 - \delta$,*

$$\mathcal{E}(\hat{w}_t) - \inf_{\mathcal{H}}\mathcal{E} \le 8\frac{\left(32\log(16/\delta)\right)^2}{n}\left[\left(M + 2M^2\kappa^{-1/2} + \frac{6\|g\|_\rho\kappa^{-1/2}}{3-2r}\right)t^{3/2-r}\right.$$

$$\left. + 3\kappa^r\frac{1-2r}{3-2r}\|g\|_\rho\right]^2 + 2\left(\frac{r}{\gamma t}\right)^{2r}\|g\|_\rho^2 \tag{84}$$

(ii) *For the stopping rule $t^*: \mathbb{N}^* \to \mathbb{N}^*$*

$$t^*(n) = \left\lceil n^{\frac{1}{3}} \right\rceil \tag{85}$$

*with probability greater than* $1 - \delta$,

$$\mathcal{E}(\hat{w}_{t^*(n)}) - \inf_{\mathcal{H}} \mathcal{E} \leq \left[ 8 \left( 32 \log \frac{16}{\delta} \right)^2 \left( M + 2M^2 \kappa^{-1/2} + \frac{6\|g\|_\rho}{3 - 2r} + \frac{3 - 6r}{3 - 2r} \kappa^r \|g\|_\rho \right)^2 \right. $$
$$\left. + 2 \left( \frac{r}{\gamma} \right)^{2r} \|g\|_\rho^2 \right] n^{-2r/3}$$

(86)

As for the attainable case, equation (84) arises from a form of bias-variance (sample-approximation) decomposition of the error. Choosing the number of epochs that optimize the bounds in (84), we derive a priori stopping rules (85) and corresponding bound (86). Again, these results confirm that the number of epochs acts as a regularization parameter and the best choice follows from equation (84).

**Proof of Theorem 7**. (i): This follows from Theorem 5(i) and Theorem 6(iii).

(ii): It follows plugging the expression of $t^*(n)$ into the inequality in (i).