[Reviews · NeurIPS 2015]

Submitted by Assigned_Reviewer_1

In the case of simple least squares regression with data from a statistical model one wants to bound and analyze the error on the true expected error rather than the empirical error on the fixed training set. This paper analyzes the expected error of a simple iterative algorithm, similar to gradient descent, which passes over the data in epochs. This setting is technically challenging as the iterates (or more precisely the gradients) are not independent of the current state due to cycling over the data. This paper gives upper bounds on both the expected error and distance of the returned parameter from the true (or best) parameter as a function of the number of epochs and as a consequence derives the optimal number of epochs based on the `niceness' of the projection of the regression function onto the relevant class of functions.

The paper is well written and has a good focus. But, it would benefit greatly from a more clear explanation of why this particular algorithm is interesting to study, (as in arguments that it is better than other algos in terms of computational complexity, robustness, practical error performance etc.)

The test error behaviour against terms of number of algorithm iterations is somewhat reminiscent of such analyses for AdaBoost (e.g. Jiang, Process consistency for AdaBoost, Annals of statistics), is there any link here?
Summary: The paper gives an insight into how running a gradient descent style algorithm for the least squares error function over multiple epochs, with the number of epochs chosen carefully as a function of the data size, can achieve optimal rates of convergence. The paper is well written and interesting.

Submitted by Assigned_Reviewer_2

The paper considers the linear squared-loss regression problems in Hilbert spaces and studies the statistical performance of particular iterative gradient-based algorithm (eq's 6 and 7) as number of epochs varies. The authors claim that if the learning rate $\gamma$ is fixed a priori, then the number of epochs acts like a regularizer. This intuition is apparently made precise with novel theoretical results on almost sure convergence of the algorithm, as well as on the finite sample risk bounds (Theorems 1, 2, 3). Finally, the authors present numerical results on the real and synthetic data sets showing, that indeed the number of epochs acts like a regularizer and should be tuned depending on a size of available data.

Overall, the paper is clearly written. The exposition is quite easy to follow. The authors also present a convincing motivation for the topic of their research. However, I feel that the discussions presented after Theorems 1,2,3 can be extended. Right now it is not quite clear how are these results related to the above mentioned intuition that the number of epochs serves a role of regularizer.

Other comments: (1) It would be nice to have a reference (or proof) for the fact that linear operator $L$ presented in (3) is self-adjoint, positive definite and trace class. These facts are well known for the RKHS. (2) Extend the discussion of Theorems 1,2,3

------

I have carefully read the authors' answers. I think they addressed all the major concerns. I am keeping my initial score.

Summary: The paper presents theoretical analysis for the role of a number of epochs in incremental learning algorithms, which is an interesting and apparently open question. The presented results look reasonable, the exposition is quite clear. There are some discussions obviously missing, but overall this is a nice and solid work.

Submitted by Assigned_Reviewer_3

Abstract

The paper considers the following scenario for least

squares regression: - the covariate X is Hilbert space H valued - the considered function class is the set of bounded

linear functionals over that Hilbert space H - the population least squares risk over H needs to be

minimized. To solve this problem a method based on the empirical

risk is considered that iteratively goes through the samples and performs a gradient step in the corresponding direction. The main theorems establish consistency (Theorem 1) and

learning rates (Theorems 2 and 3) under early stopping.

These results are both in terms of excess risk and

H-norm (if a minimizer exists).

Comments

The paper nicely describes the regularization effect of early stopping for this particular learning problem. In addition, the considered rather general setup

includes both linear regression and regression with RKHSs (for the latter it would have been nice though, to

include a computational cost estimate). On the downside, the obtained rates for excess risk are far from being

optimal. The proofs, which are mostly deferred to an appendix, use Hilbert space valued concentration inequalities, but the way the proof is set up, it seems that significant

new work is needed compared to the case of Thikonov

regularization ("kernel least squares regression" or

"least squares regression").

Here some minor comments:

pages 2. It should be better communicated why the

Hilbert space approach is taken (instead of an RKHS approach). Also, it should be better explained how the latter is recovered from your general setu.

page 3, line following (4). I have no idea where an assumption like (3) and (4) can be found in [28]. This needs to be explained in detail.
Summary: The paper investigates an iteration scheme for minimizing

the empirical least squares loss over a Hilbert space. The main results are statistical consistency and learning rates under some early stopping regimes. A very few

experiments are also reported.

Submitted by Assigned_Reviewer_4

The paper addresses an interesting high-level question: How does the number of iterations in a stochastic gradient method affect the generalization properties of the learned model?

This work addresses the general problem in the setting of least squares minimization using a variant of stochastic gradient descent. The algorithm works in epochs each of which corresponds to one sequential pass over the data.

Rather than using an explicit regularization term in the objective functions, the main result achieves consistency and convergence of the empirical risk through a carefully chosen "early" stopping rule.

I find the general direction of this work important and the paper makes a solid technical contribution. As I was asked for a "light" review, I did not have time to venture into the dense technical parts. For the NIPS conference format, it would definitely help if the authors highlighted the simplest novel corrolaries of their work rather than presenting several general theorems. For example, rather than jumping straight into the setting of an abstract Hilbert space, what can you say for the basic Euclidean setting?
Summary: A solid technical contribution to an interesting research direction. The paper would benefit from a more accessible presentation.

Author Feedback
Author rebuttal: We thank the reviewers for
all the comments and suggestions. We reply here to the main points raised in the reviews.
Rev. 1
The main motivation of our work comes from the fact that doing multiple passes over
the data is a very common techinque in machine learning, used also when stochastic gradient method is adopted (and in principle one pass would suffices). Indeed, this usually leads to better practical results. We will better clarifies this in the introduction.
Moreover, we will add some more discussion after the main theorems to better explain the advantages of the proposed method with respect to other approches, especially
in terms of computational complexity and stability.
The results in the mentioned paper on Adaboost are similar in spirit, showing that early stopping leads to consistent estimators. However, it seems to us that the assumptions and the proof techniques are quite different. We will add this related reference in the paper.
Rev. 2
We will enrich and deepen the discussion following the main results (see also the answer to rev. 1)
We will add the reference for the properties of the operator L.
Rev. 3
We chose the HIlbert space approach because it immediately generalizes the finite dimensional linear case. The derivation of the RKHS setting is proposed in the appendix, and we will add omitted details to make it clearer.
We will add precise references when needed (equations (3) and (4)).
Rev. 4
See the answer to Rev. 1, Rev.2 and Rev. 3 about the discussion following the main results and the Hilbert space approach.
Rev. 5
We think that Remark 1 is correct, and we will make it clearer to avoid confusion. There, we refer to the case where a single epoch is taken, and the given points are iid. This is the setting of stochastic gradient method. Any choice of the order of the points works in the analysis (simply by renaming the samples). We agree with the reviewer that gamma/n obviously depend on n, but while in the multiple epochs case, gamma can be choosen as a fixed constant independent on the parameters of the problem (the smoothness parameter r and the number of points n),
in the single epoch must be chosen depending on n and r.